# Mechanical Properties of ASTM A572 Grades 50 and 60 Steels at High Temperatures

Su-Hyeon Lee [ID] and Byong-Jeong Choi *[ID]

Department of Architectural Engineering, Kyonggi University, Suwon 16227, Korea; hyeon1284@naver.com
* Correspondence: bjchoi@kgu.ac.kr

**Abstract:** Studies involving the mechanical properties of high-strength steel (HSS) at elevated temperatures have received considerable attention in recent years. However, current research on HSS at high temperatures is lacking. As a result, the design of fire-protective steel structures with high standards is not sufficiently conservative or safe. This study investigates the effect that elevated temperatures have on the mechanical properties of ASTM A572 Gr. 50 and 60 steels. Reduction factors for the yield strength, tensile strength, and elastic modulus were derived and compared with the standard (AISC, EN1993-1-2) and previous studies (NIST). This study also provides extensive data on the reduction factors for the yield strength, tensile strength, and elastic modulus of mild steel (MS), HSS, and very-high-strength steel (VHSS). The reduction factor for the yield strength was analyzed by expanding the strain level up to 20%. Equations for the yield strength, tensile strength, and elastic modulus were proposed. In future studies, various strains should be analyzed according to the grade of the steel, with the derivation of a reduction factor that considers the plastic strain of the steel. Hence, the findings reported in this study generated a database that can be applied to fire safety design or performance-based fire-resistant design.

**Keywords:** ASTM A572 steel; high-strength steel; elevated temperatures; steady state test; stress–strain curves with temperature; strain level; mechanical properties; reduction factor



## 1. Introduction

An elevated temperature of a structure exposed to fire can cause temperature-dependent effects on the building materials, such as concrete and steel [1]. Generally, a structure can experience large deflection with the reduction in both strength and stiffness at elevated temperatures, as mentioned in Chapter 1, MOP, ASCE [1]. Therefore, it is necessary to know the mechanical properties, including the Young's modulus reduction, yield strength, and tensile strength reduction, under various stress levels at elevated temperatures. The stress–strain relationships must be acquired to draw out the mechanical properties at elevated temperatures. Both steady-state and transient-state tests are being used to induce the mechanical properties at high temperatures currently. As we know, the steady-state test has been widely used to evaluate the stress–strain relationship of steel members because of its simplicity and practicality [1–3]. The steady-state tests provide the stress–strain results directly [4]; however, the transient-state tests result in a temperature–strain curve that needs to be modified into a stress–strain curve. In a transient-state test, a series of works are required to convert a temperature–strain curve into a stress–strain curve. There is the possibility of data being missed or differing due to the modification of the analytical processes on the results [2]. Thus, many of the research results regarding the stress–strain curve and strength reduction factors at elevated temperatures are reliant on the steady-state test. Most of the steady-state tests disregard the creep effects for the following reasons. First, the creep effects are offset due to the heating of the steel member at the jig setting for a short period of ten minutes or for a long period of more than one hundred minutes (because both ends of the steel are fixed by the jig). Secondly, data differences can easily arise not

from the heating rate but from the loading rate [5,6]. In other words, the steady-state test implicitly considers creep effects through loading rates under constant-temperature conditions [1]. Thirdly, the creep effect is disregarded during the tension test with a short period of time of approximately two to three minutes at elevated temperatures [7,8]. If it is necessary to determine the creep effect, a separate creep test is needed [8]. The creep effect also implicitly is considered in EN 1993-1-2 [9] in the steel material [1,4]. In this paper, the steady-state test was selected and carried out because the purpose of the research was to provide the stress–strain curve and determine the difference in the mechanical properties of mild steel (MS) and high-strength steel (HSS) under high-temperature conditions.

Among the various steel materials in building construction, the high-strength low alloy (HSLA) provides better strength, weldability, corrosion, and weather resistance compared to conventional steel materials [10]. Most of the HSLA are ASTM A572 Grade 50 (Gr. 50) and have the same characteristics as the HSLA 355 in Europe. Because Gr. 50 steel is increasingly used in buildings and bridges, it is increasingly important to know its thermal properties. There are many existing studies on the Gr. 50 steel; however, very few are available on the ASTM A 572 Grade 60 (Gr. 60) in comparison with the Gr. 50. As mentioned earlier, additional studies on high-strength steel (Gr. 60) are necessary at elevated temperatures in accordance with the increase in demand. It is called high-strength steel (HSS) if the normal yield strength exceeds 460 N/mm$^2$ based on the EN 1993-1-2 [9]. The ASCE MOP [1] also states that not only is the yield strength of high-strength steel (HSS) between 490 N/mm$^2$ and 690 N/mm$^2$ but also the yield strength of very-high-strength steel (VHSS) is more than 690 N/mm$^2$. In the paper, the strength categories are presented in Table 1 for various steel materials, including mild steel, HSS, and VHSS, for convenience [11,12]. The recent research can be summarized as follows.

**Table 1.** Types and applications of steel according to the yield strength.

| Yield Strength (MPa) | Type | Description | Typical Example | Application |
|---|---|---|---|---|
| <400 | Mild steel (MS) | Regular structural steel | SS235 (SS400), SM355 (SM490), A572 (Gr. 50), S350, S355, A992 | Buildings |
| 400–690 | High-strength steel (HSS) | High-performance steel/high tensile steel | A572 (Gr. 60), S460 | Bridges, high-rise buildings |
| >690 | Very-high-strength steel (VHSS) | Ultra/super-high-strength steel | S690, S960 | Cranes, bridges, high-rise buildings, offshore structures |

Chen et al. [13] examined the reduction factors for the yield strength, elastic modulus, and ultimate strength using high-strength structural steel. They also covered the thermal characteristics of MS as compared with HSS. They showed that the thermal characteristics of HSS are different from those of MS. However, there is no presented regression equation for strength reduction depending on high-strength steel and mild steel by temperature.

Qiang et al. [14] investigated HSS using S690 in a fire scenario. They verified the thermal properties of HSS using both steady- and transient-state tests. The results of the HSS S690 test were compared with the AISC (American), ASCE (American), Eurocode 3 (European), and AS 4100 (Australian) standards. As a result of the comparison, it was found that the current design standard cannot be safely applied to the fire-resistant design of steel structures using HSS S690. The study of Qiang derives significant data on mechanical properties at high temperatures, as in the study of Chen et al. [13], but the regression equation for HSS S690 was not presented. In addition, the results and differences for high-strength steel compared to mild steel were not described.

Aziz et al. [15] also evaluated the residual strength of a fire-exposed steel bridge girder. In their FEM modeling studies, a case study of a steel bridge girder showed a residual capacity of ~84% at a fire temperature of 680 °C. In order to check the thermal and structural response of the steel bridge girder exposed to fire, the thermal and mechani-

cal properties of the steel were entered into a finite element analysis program (ANSYS). The high-temperature mechanical properties of steel are useful for FEM analysis.

Qiang et al. [16] obtained the post-fire elastic modulus, yield and ultimate strength reductions, ductility reductions, and stress–strain curves of HSS (S460, S690). Their results showed that the steel grade has a significant influence on the post-fire residual mechanical properties of HSS. They found that the post-fire mechanical properties of S460 and S690 are not affected until they are exposed to temperatures above 600 °C.

Wang et al. [17] presented results from experimental studies on the high-temperature properties of high-strength Q460 steel. Based on their results, they concluded that high-strength Q460 steel exhibits a slower loss of strength and stiffness than MS throughout a temperature range of 20–800 °C. Moreover, Wang proposed the elastic modulus equation for high-strength steel Q460, but there was a difference from Eurocode 3 based on mild steel. This study contained significant results on the elastic modulus of Q460 steel, but the detailed results and related equations for the difference between mild steel and high-strength steel were somewhat insufficient.

Lee et al. [18] estimated the stress–strain of ASTM A992 steel at elevated temperatures from 20 to 1000 °C with strain regions at both 5% and 20%. ASTM A992 steel has a specified minimum yield stress of 345 MPa (50 ksi) and specified maximum yield stress of 450 MPa (65 ksi). The estimation of the stress–strain relation with their detailed model showed results more similar to EC3 and NIST. Lee et al. [19] also pointed out that the cooling methods can affect the residual strength after a fire.

Kodur et al. [10] examined the effect of temperature on creep deformations in high-strength, low-alloy ASTM A572 steel. They suggested that creep deformations of A572 steel are not significant until 500 °C and become predominant at temperatures above 500 °C. They also specified the creep effect on steel members, where creep deformation becomes predominant at temperatures above 500 °C.

Azhari et al. [20] investigated the reduction in the strength of ultra-high-strength (grade 1200) steel tubes after cooling from fire temperatures of up to 600 °C, which does not occur to the same extent for HSS and MS. HSS did not experience significant strength reductions after cooling from fire temperatures of up to 600 °C. The difference in residual strength after cooling of MS, HSS, and UHSS was derived through microstructure observation using SEM.

Aziz et al. [21] tested ASTM A572 Gr. 50 steel to obtain mechanical property data, including tensile strength (stress–strain response during heating) and residual strength (residual stress–strain response after cooling) evaluations at various temperatures. If A572 steel is cooled after heating up to 600 °C, it recovers almost 100% of its yield strength at room temperature. The equation of ASTM A572 Gr. 50 was proposed and compared with A992 and S690 steel. However, this study did not provide a comparison with ASTM A572 Gr. 60 steel, and a comparative analysis for mild steel and high-strength steel was not conducted.

Maraveas et al. [12] analyzed the mechanical properties of HSS and VHSS at elevated temperatures and after the cooling state. They also discussed the fact that the most recent codes have conservatively reported on the reductions in the yield strength for MS; however, this is not the case for HSS.

Qiang et al. [22] examined the mechanical properties of high-strength structural steel (S460N) at elevated temperatures. The recommendations for the mechanical properties of structural steel for current European and American steels were mainly obtained from MS. Compared with the current European, American, Australian, and British steel structure design standards, it was deduced that high-strength SN460N steel is not safe for fire-resistant design. Qiang's study emphasizes the need for more research on the mechanical properties at high temperatures of all high-strength steel grades used in construction.

Li et al. [23] also analyzed the post-fire mechanical properties of high-strength Q690 structural steel. Their study found that high-temperature treatments and cooling methods can significantly affect the yield strength, ultimate strength, and elongation at 500 °C. A review of the literature indicates that there is a lack of information on HSS at elevated temperatures.

Most tests have reported that the mechanical properties are different from those of conventional carbon steels [24,25]. Moreover, the mechanical properties are mentioned in the MOP Sec.8.2.1.2, ASCE MOP 138 in terms of stress–strain relationships for structural steel at elevated temperatures [1]. It is mentioned that the most important mechanical properties of structural steel all decrease as the temperature increases. Nevertheless, it is not mentioned whether the types of structural steel are mild steel or high-strength steel.

In this research, we focus on the difference in the strength reduction factor between mild steel (MS) and high-strength steel (HSS). In addition, ASCE MOP states that caution should be exercised because high-strength steels may experience greater strength loss at elevated temperatures than mild steels [1].

There are not many mechanical property datasets specific to both the ASTM A572 Gr. 50 and 60 steels for fire design purposes. In addition, most of the studies provided only the data sheet of the tested steel, and there were few comparison targets with other steel types. Since the mechanical properties are different according to the steel type, it is necessary to analyze the data and propose various expected equations for each type of steel. Therefore, previous test results were analyzed to propose various reduction factors for mild steel (MS) and high-strength steel (HSS), and additional tests were conducted to compare these to proposed results groups using ASTM A572 Gr. 50 and 60 steels.

Table 2 lists the yield strength and tensile strength values of each study that examined the stress–strain relationship at elevated temperatures. Figures 1 and 2 summarize the results of the yield strength reduction factor and tensile strength reduction factor for MS, HSS, and VHSS. First, Figure 1 shows the MS, HSS, and VHSS yield strength reduction factor graph. Despite using the same steel type, there was a difference in the yield strength reduction factor, as shown in Figure 1a, according to the strain level. In particular, the classification of MS, HSS, and VHSS according to steel type is prominent in Figure 1b. EN1993-1-2 (blue fine line) and AISC_2016 (black dotted-dashed fine line) are drawn in the MS zone. In contrast, the regression equation represented by NIST (NIST_2016, green dashed fine line) is drawn in the HSS zone up to 450 °C and in the MS zone above 450 °C. There is not a significant amount of data on VHSS, but it is distributed below in the MS and HSS regions based on S960. As the temperature increased, the strength of HSS and VHSS decreased sharply compared with MS, as shown in Figure 1. These analyses indicate that the yield strength reduction factor depends on the steel type, test method, and strain level. In general, the Eurocode and AISC propose a strength reduction factor in the MS range. As the temperature increases, the yield strength reduction of HSS and VHSS is greater than that of MS, such that the strength reduction factor of HSS and VHSS at high temperatures should be applied more conservatively than the standard. Therefore, the yield strength reduction factor should be separately presented according to the steel type (MS, HSS, and VHSS). In this study, a yield strength reduction factor formula for MS and HSS is proposed. Figure 2 shows the MS, HSS, and VHSS tensile strength reduction factor graph. The tensile strength reduction factor indicates the maximum value of the yield strength regardless of the strain level. As shown in Figure 2b, EN1993-1-2 (blue fine line) shows a conservative tensile strength reduction factor compared with other studies, but AISC (black dotted-dashed line) was relatively overestimated. In addition, the tensile strength reduction factor may have a value greater than 1.0 at 100–300 °C, which is a phenomenon due to temperature hardening and has been reported in several studies [22]. Therefore, more research is required to apply and analyze the tensile strength reduction factor above HSS to the design. These figures show that the reduction in the yield and tensile strength of HSS decreases compared with MS. Thus, engineers must be careful when using the strength reduction for HSS at temperatures between 500 °C and 600 °C. Figure 3 summarizes the reduction factors for the elastic modulus of previous tests for MS, HSS, and VHSS. The elastic modulus reduction factors are also reduced further than those for MS at elevated temperatures. Based on the results of previous studies, the strength reduction in HSS must be investigated further for fire design purposes. To fill this knowledge gap, a series of experimental studies on the mechanical characteristics at elevated

temperatures are performed for ASTM A572 steels with the objective of generating the strength reduction factors at various strain levels. These experimental studies with the steady-state test included the stress–strain relationships at strains of 0.05, 0.1, and 0.2% for both the A572 Gr. 50 and 60 steels. Results from the stress–strain relationship at elevated temperatures are used to suggest the strength reduction factors with convenient regression equations for both the A572 Gr. 50 and 60 steels. To compare the mechanical properties of A572 Gr. 60 to those of mild steel, Gr. 50 steel was selected in this study to quantify and compare the above properties.

**Table 2.** Types of steel according to yield strength.

| Group | Type | Grade | Yield Strength $\left(f_y, \mathrm{MPa}\right)$ | Tensile Strength $(f_u, \mathrm{MPa})$ | Author |
|---|---|---|---|---|---|
| Index | EN1993-1-2 | Eurocode Standard (EN1993-1-2) [9] | | | |
| | NIST_2016 | NIST Regression Equation [26] | | | |
| | AISC_2016 | AISC Standard [27] | | | |
| MS | SS400_1 | SS400 (SS235) | 235 | 400 | Kwon (2001) |
| | SS400_2 | | | | Kwon (2007) |
| | SM490 | SM490 (SM355) | 355 | 490 | Kwon (2007) |
| | A572 Gr. 50_Kodur | ASTM A572 Grade 50 | 345 | 450 | Kodur (2016) |
| | A572 Gr. 50_Test | | | | This study |
| | A992 | ASTM A992 | 345 | 450 | Lee (2015) |
| | S350GD+Z | S350GD+Z | 350 | 420 | |
| | S355J2H_50×50×3 | S355J2H | 355 | 470–630 | Outinen (2004) |
| | S355J2H_80×80×3 | | | | |
| | S355J2H_100×100×3 | | | | |
| | Mild steel | Mild steel | 400–460 | 550 | Chen (2006) |
| HSS | A572 Gr.60_Test | ASTM A572 Grade 60 | 415 (Min) | 520 | This study |
| | S460N_SST | S460N | 460 | 540–720 | Qiang (2013) |
| | S460N_TST | | | | |
| VHSS | S690_SST | S690 | 690 | 770–940 | Qiang (2013) |
| | S690_TST | | | | |
| | HSS | High-strength steel | 780–820 | 850 | Chen (2006) |
| | S960_SST | S960 | 960 | 980–1150 | Qiang (2013) |
| | S960_TST | | | | |

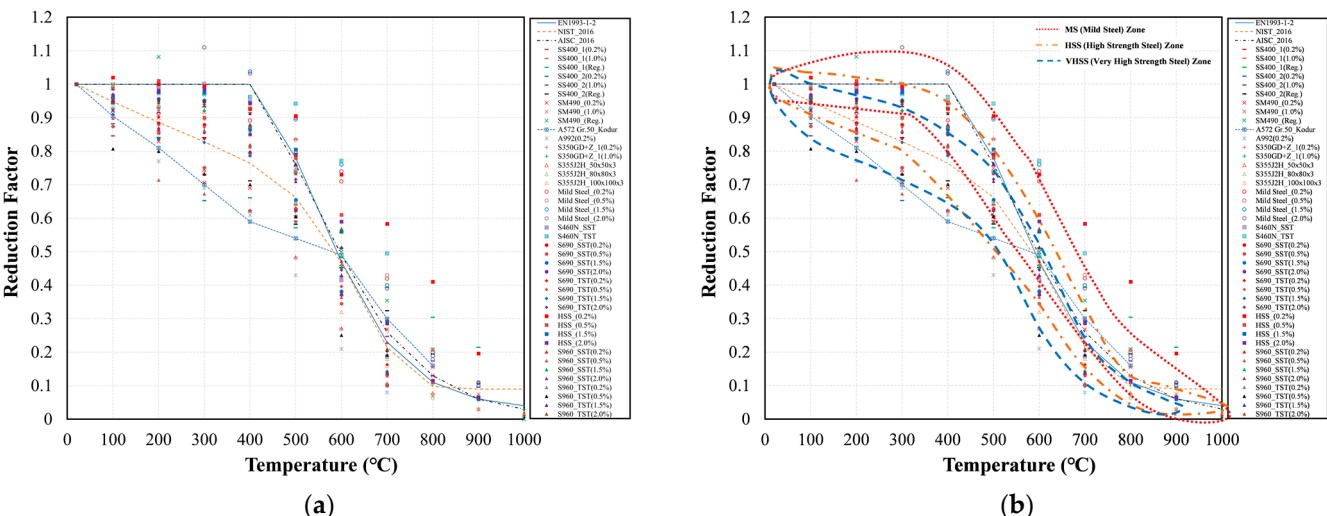

**Figure 1.** The yield strength reduction factor: (**a**) different steels at elevated temperatures from previous studies; (**b**) according to the steel grade, highlighting the MS, HSS, and VHSS zones.

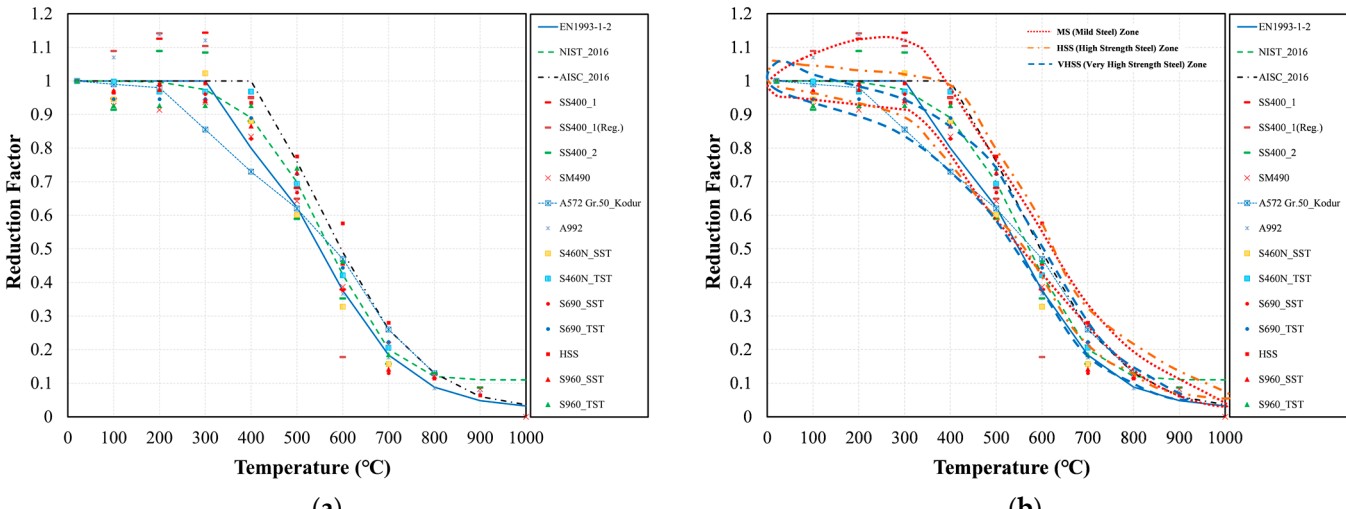

**Figure 2.** The tensile strength (ultimate strength) reduction factor: (**a**) different steels at elevated temperatures from previous studies; (**b**) according to the steel grade, highlighting the MS, HSS, and VHSS zones.

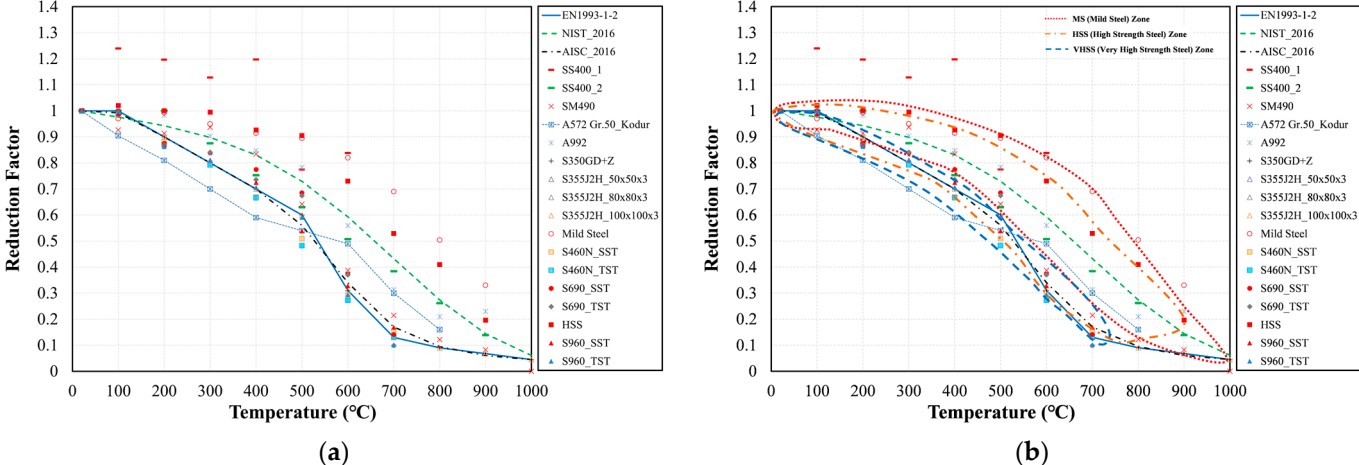

**Figure 3.** The elastic modulus reduction factor: (**a**) different steels at elevated temperatures from previous studies; (**b**) according to the steel grade, highlighting the MS, HSS, and VHSS zones.

## 2. Experimental Tests

### 2.1. Test Specimen Properties

To investigate the reduction factors of the yield strength, tensile strength, and elastic modulus, 66 samples were prepared for both the A572 Grades 50 (Gr. 50) and 60 (Gr. 60) steels. Table 3 lists the number of samples for both steels, where the plate thickness for the test sample was 6 mm. The target temperatures were 25 °C (ambient temperature), 100 °C, and 900 °C, where the temperature was increased at intervals of 100 °C. Figure 4 shows the shapes of the samples and their dimensions. The rectangular steel plate was laser-cut for the high-temperature tensile test of steel. The total length (TL) of each steel coupon was 910 mm, and the length of the tapered section (P) was 70 mm. The reduced section length (L) for gauge measurement was 50 mm, the reduced section was 20 × 6 mm, the width was 20 mm, and the thickness (T) was 6 mm. The fillet radius (R) was 15 mm. Table 4 lists the mechanical properties of the A572 Gr. 50 and 60 steels. The yield strengths of Gr. 50 and 60 were 345 and 415 MPa, respectively. In this study, Gr. 60 steel was considered to be closely grouped with HSS. The Gr. 50 steel was selected to compare the strength reduction in MS against that in HSS. A yield strength of 0.2% offset was used and the stress–strain relationship is shown in Figure 5. The chemical composition of ASTM A572 Gr. 50 steel and Gr 60 steel is summarized in Table A1 of the Appendix A.

**Table 3.** Summary of the steady-state test performed at elevated temperatures.

| Test | Steel Type | Plate Thickness (mm) | Target Temperature (°C) | | | | | | | | | | |
|------|-----------|------|------|-----|-----|-----|-----|-----|-----|-----|-----|-----|------|
| | | | 25 * | 100 | 200 | 300 | 400 | 500 | 600 | 700 | 800 | 900 | 1000 |
| SST | ASTM A572 Grade 50 (Gr. 50) | 6 | 3 ** | 3 | 3 | 3 | 3 | 3 | 3 | 3 | 3 | 3 | 3 |
| | ASTM A572 Grade 60 (Gr. 60) | 6 | 3 | 3 | 3 | 3 | 3 | 3 | 3 | 3 | 3 | 3 | 3 |

\* Ambient temperature, \*\* three specimens per test

**Table 4.** Mechanical Properties of ASTM A572 Gr. 50 and 60 steels.

| Mechanical Properties | Steel Type | |
|-----------------------|----------------|----------------|
| | **ASTM A572 Gr. 50** | **ASTM A572 Gr. 60** |
| Yield Strength (MPa) | 345 | 415 |
| Tensile Strength (MPa) | 450 | 520 |
| Elastic Modulus (MPa) | 200,000 | 200,000 |
| Elongation Break (%) | 18–21 | 16–18 |
| Bulk Modulus (GPa) | 160 | 160 |
| Shear Modulus (GPa) | 80 | 80 |

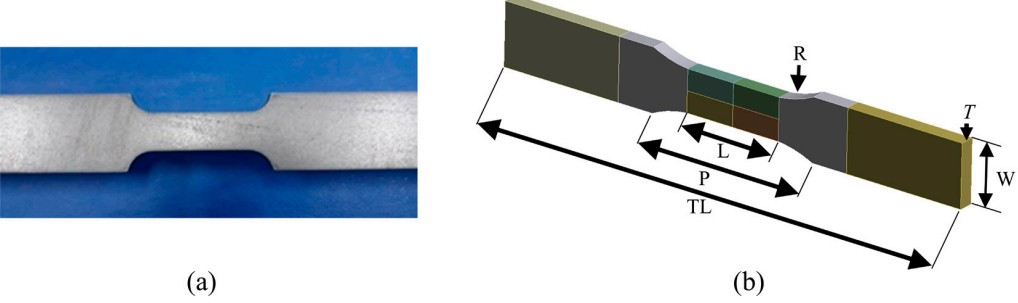

(a)          (b)

**Figure 4.** Steel specimen: (**a**) steel coupon; (**b**) steel coupon dimensions.

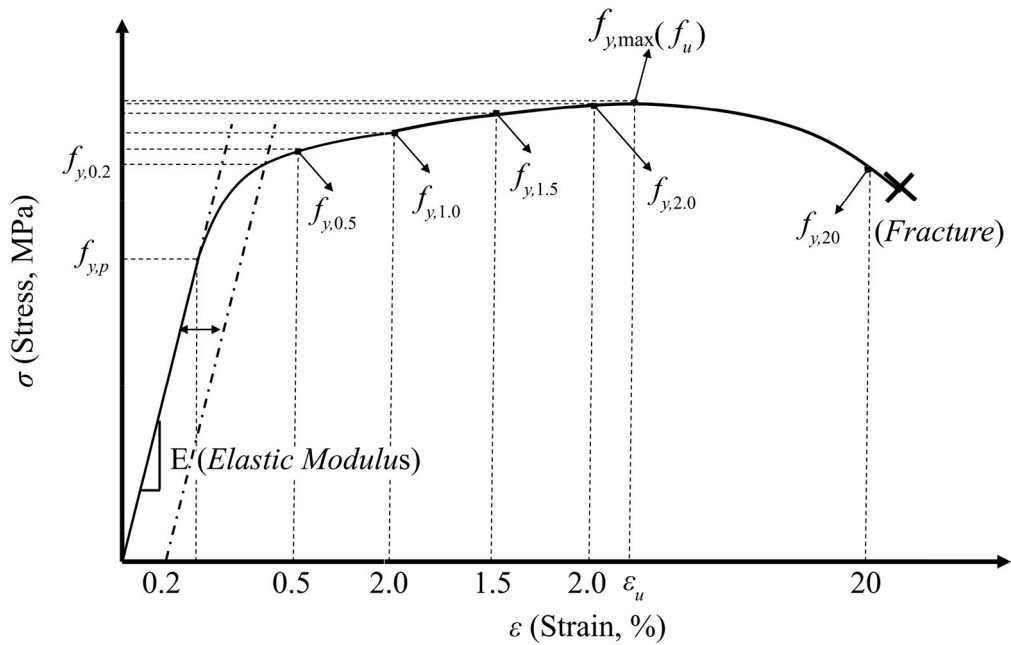

**Figure 5.** Determination of the yield strength, tensile strength, and elastic modulus according to strain level.

### 2.2. Test Methods for Thermal Properties

The tests were performed in accordance with ASTM E8 [28] and ASTM E21 [29]. The two methods are both steady-state and transient-state tests at elevated temperatures for steel coupons. In this study, the steady-state test method was used to investigate the stress–strain behavior at elevated temperatures.

In the steady-state test, the samples were heated to a specified target temperature and then loaded until they failed while maintaining the same temperature. To measure the tensile strength of steel at high temperature, the experimental equipment setting consisted of a steel frame, an electric furnace, a high-temperature extensometer, and data logger equipment. The load cell of the universal testing machine (UTM) had a loading capacity of ±2000 kN, where the displacement transducer could measure up to 250 mm. An electric heating furnace to surround a steel specimen can be heated up to 1200 °C and maintain a constant temperature for an extended period. A thermocouple was installed inside the electric heating furnace to verify the internal temperature in real time during heating. The high-temperature extensometer used the Epsilon Model 3549 (Epsilon Technology Co., Ltd., WY, USA), which can measure gauge lengths of up to 50 mm. This high-temperature extensometer, with a sensitivity of 0.00694 mm for axial deformation, was measured by a data logger device, i.e., TDS-150 (Tokyo Sokki Kenkyujo Co., Ltd., Tokyo, Japan). The load displacement data for the UTM, temperature of the electric heating furnace, and displacement measurement data for the high-temperature extensometer were connected to the data logger device and transmitted to the computer. In addition, not only the temperature of the thermocouple inside the heating furnace was measured, but also the surface temperature of the steel was separately measured using SDT25 (Type K Thermocouple) equipment (Figure A1). The samples were heated from ambient temperature to a target temperature, i.e., 25, 100, 200, 300, 400, 500, 600, 700, 800, and 900 °C. The heating rate of the steel specimen was 10 °C/min. After reaching the target temperature, a general 30 min period was required for temperature stabilization; after 15 min, the tensile load was applied to the samples. In the steady-state test, strain control was achieved with the UTM. The strain rate was ~0.006/min, which is within the range of 0.005 ± 0.002/min established by ASTM E21-92 (1997) [30].

## 3. Results and Discussion

The mechanical properties of ASTM A572 steel at high temperatures are useful for deriving the stress–strain relationship. This section presents the tensile test results for each target temperature for ASTM A572 steel. The presented data are compared with the values reported in codes and previous studies. In addition, regression equations are presented based on the data analysis. In Tables 5–8, the 0.2% offset ($f_{y,0.2}$) is the high-temperature yield strength at a strain level of 0.2%, while $f_{y,0.5}$, $f_{y,1.0}$, $f_{y,1.5}$, $f_{y,2.0}$, and $f_{y,20}$ indicate the high-temperature yield strength at strain levels of 0.5, 1.0, 1.5, 2.0, and 20%, respectively. Here, $f_{y,25,0.2}$, $f_{y,25,0.5}$, $f_{y,25,1.0}$, $f_{y,25,1.5}$, $f_{y,25,2.0}$, and $f_{y,25,20}$ are the yield strengths at strain levels of 0.2, 0.5, 1.0, 1.5, 2.0, and 20%, respectively, at room temperature. $f_u$ represents the highest strength value at all strain levels and $f_{u,25}$ is the ultimate strength at room temperature (25 °C). $E_T$ represents the high-temperature elastic modulus, whereas $E_{T,25}$ is the elastic modulus at room temperature. The strength derived at each target temperature was normalized by the strength at room temperature and expressed as a reduction factor (RF).

**Table 5.** Strength values and elastic moduli of Gr. 50 at various strain levels obtained from the steady-state test.

| Target Temperature (°C) | Strength Value (MPa) | | | | | | | Elastic Modulus (MPa) |
|---|---|---|---|---|---|---|---|---|
| | 0.2% ($f_{y,0.2}$) | 0.5% ($f_{y,0.5}$) | 1.0% ($f_{y,1.0}$) | 1.5% ($f_{y,1.5}$) | 2.0% ($f_{y,2.0}$) | 20% ($f_{y,20}$) | Max ($f_u$) | $E_T$ |
| 25 | 369 ($f_{y,25,0.2}$) | 366 ($f_{y,25,0.5}$) | 385 ($f_{y,25,1.0}$) | 390 ($f_{y,25,1.5}$) | 406 ($f_{y,25,2.0}$) | 501 ($f_{y,25,20}$) | 508 ($f_{u,25}$) | 206,666 ($E_{T,25}$) |
| 100 | 353.33 | 356 | 368 | 386.7 | 402.7 | 460 | 484 | 184,000 |
| 200 | 321.5 | 322.17 | 341.24 | 351.05 | 366.7 | 481.5 | 481.71 | 225,000 |
| 300 | 314 | 314.6 | 352 | 372 | 384 | 464 | 477.3 | 166,667 |
| 400 | 278.7 | 276 | 305.3 | 317.3 | 329.3 | 334.7 | 368 | 170,000 |
| 500 | 217.3 | 221.3 | 237.3 | 242.7 | 248 | 228 | 257.3 | 105,000 |
| 600 | 136 | 144 | 150.7 | 153.3 | 154.7 | 122.7 | 154.7 | 81,666 |
| 700 | 89.4 | 92 | 90.7 | 92 | 92 | 74.7 | 92 | 55,000 |
| 800 | 50 | 52 | 52 | 52 | 54.6 | 50 | 56 | 25,000 |
| 900 | 36 | 38.7 | 41.3 | 42.7 | 42.7 | 38.7 | 48 | 21,667 |

**Table 6.** Reduction factors for Gr. 50 at various strain levels obtained from the steady-state test.

| Target Temperature (°C) | Strength Reduction Factor (RF) | | | | | | | Elastic Modulus RF |
|---|---|---|---|---|---|---|---|---|
| | 0.2% ($f_{y,0.2}/f_{y,25,0.2}$) | 0.5% ($f_{y,0.5}/f_{y,25,0.5}$) | 1.0% ($f_{y,1.0}/f_{y,25,1.0}$) | 1.5% ($f_{y,1.5}/f_{y,25,1.5}$) | 2.0% ($f_{y,2.0}/f_{y,25,2.0}$) | 20% ($f_{y,20}/f_{y,25,20}$) | Max ($f_u/f_{u,25}$) | $E_T/E_{T,25}$ |
| 25 | 1.00 | 1.00 | 1.00 | 1.00 | 1.00 | 1.00 | 1.00 | 1.00 |
| 100 | 0.96 | 0.97 | 0.96 | 0.99 | 0.99 | 0.92 | 0.95 | 0.89 |
| 200 | 0.87 | 0.88 | 0.89 | 0.90 | 0.90 | 0.96 | 0.95 | 1.09 |
| 300 | 0.85 | 0.86 | 0.91 | 0.95 | 0.95 | 0.93 | 0.94 | 0.81 |
| 400 | 0.76 | 0.75 | 0.79 | 0.81 | 0.81 | 0.67 | 0.72 | 0.82 |
| 500 | 0.59 | 0.60 | 0.62 | 0.62 | 0.61 | 0.46 | 0.51 | 0.51 |
| 600 | 0.37 | 0.39 | 0.39 | 0.39 | 0.38 | 0.24 | 0.30 | 0.40 |
| 700 | 0.24 | 0.25 | 0.24 | 0.24 | 0.23 | 0.15 | 0.18 | 0.27 |
| 800 | 0.14 | 0.14 | 0.14 | 0.13 | 0.13 | 0.10 | 0.11 | 0.12 |
| 900 | 0.10 | 0.11 | 0.11 | 0.11 | 0.11 | 0.08 | 0.09 | 0.10 |

**Table 7.** Strength values and elastic moduli for Gr. 60 at various strain levels obtained from the steady-state test.

| Target Temperature (°C) | Strength Value (MPa) | | | | | | | Elastic Modulus (MPa) |
|---|---|---|---|---|---|---|---|---|
| | 0.2% ($f_{y,0.2}$) | 0.5% ($f_{y,0.5}$) | 1.0% ($f_{y,1.0}$) | 1.5% ($f_{y,1.5}$) | 2.0% ($f_{y,2.0}$) | 20% ($f_{y,20}$) | Max ($f_u$) | $E_T$ |
| 25 | 332 ($f_{y,25,0.2}$) | 292 ($f_{y,25,0.5}$) | 358.7 ($f_{y,25,1.0}$) | 397.3 ($f_{y,25,1.5}$) | 418.7 ($f_{y,25,2.0}$) | 466.7 ($f_{y,25,20}$) | 514.7 ($f_{u,25}$) | 205,000 ($E_{T,25}$) |
| 100 | 310.7 | 306.7 | 360 | 400 | 425.3 | 476.85 | 496 | 180,000 |
| 200 | 321.3 | 318.7 | 373.3 | 412 | 437.7 | 502.65 | 516 | 182,000 |
| 300 | 298.6 | 316 | 369.3 | 406.7 | 437.3 | 502.65 | 568 | 248,000 |
| 400 | 268 | 276 | 320 | 350.7 | 374.7 | 405.3 | 460 | 220,000 |
| 500 | 196.7 | 201.3 | 229.3 | 246.7 | 257.3 | 205.3 | 273.3 | 116,667 |
| 600 | 128 | 138.7 | 152 | 156 | 160 | 133.3 | 162.7 | 85,000 |
| 700 | 60 | 72 | 76 | 77.7 | 77.7 | 58.7 | 77.7 | 50,000 |
| 800 | 41 | 42.7 | 45.3 | 50.7 | 52 | 52 | 61.3 | 25,000 |
| 900 | 28 | 28 | 32 | 33.3 | 33.3 | 34.7 | 40 | 16,667 |

**Table 8.** Reduction factors for Gr. 60 at various strain levels obtained from the steady-state test.

| Target Temperature (°C) | Strength Reduction Factor (RF) | | | | | | | Elastic Modulus RF |
|---|---|---|---|---|---|---|---|---|
| | 0.2% ($f_{y,0.2}/f_{y,25,0.2}$) | 0.5% ($f_{y,0.5}/f_{y,25,0.5}$) | 1.0% ($f_{y,1.0}/f_{y,25,1.0}$) | 1.5% ($f_{y,1.5}/f_{y,25,1.5}$) | 2.0% ($f_{y,2.0}/f_{y,25,2.0}$) | 20% ($f_{y,20}/f_{y,25,20}$) | Max ($f_u/f_{u,25}$) | $E_T/E_{T,25}$ |
| 25 | 1.00 | 1.00 | 1.00 | 1.00 | 1.00 | 1.00 | 1.00 | 1.00 |
| 100 | 0.94 | 1.05 | 1.00 | 1.01 | 1.02 | 0.99 | 0.96 | 0.88 |
| 200 | 0.97 | 1.09 | 1.04 | 1.04 | 1.05 | 1.02 | 1.00 | 0.89 |
| 300 | 0.90 | 1.08 | 1.03 | 1.02 | 1.04 | 1.08 | 1.10 | 1.21 |
| 400 | 0.81 | 0.95 | 0.89 | 0.88 | 0.89 | 0.87 | 0.89 | 1.07 |
| 500 | 0.59 | 0.69 | 0.64 | 0.62 | 0.61 | 0.44 | 0.53 | 0.57 |
| 600 | 0.39 | 0.48 | 0.42 | 0.39 | 0.38 | 0.29 | 0.32 | 0.41 |
| 700 | 0.18 | 0.25 | 0.21 | 0.20 | 0.19 | 0.13 | 0.15 | 0.24 |
| 800 | 0.12 | 0.15 | 0.13 | 0.13 | 0.12 | 0.11 | 0.12 | 0.12 |
| 900 | 0.08 | 0.10 | 0.09 | 0.08 | 0.08 | 0.07 | 0.08 | 0.08 |

### 3.1. Mechanical Properties

#### 3.1.1. Stress–Strain Curve

After testing the ASTM A572 Gr. 50 steel, the stress–strain curves at each target temperature were summarized, as shown in Figure 6. Figure 6a shows the stress–strain curve up to the initial 0.05 (5%) strain level. Using the stress–strain curve at the initial strain level, the diagram of the steel in the elastic section can be carefully observed. Figure 6b shows the stress–strain curve up to the 0.2 (20%) strain level. In general, it is common to obtain the strain of steel for design purposes of up to 20%; in this study, the range of the maximum strain was expanded to 20%. Based on the stress–strain curve of Gr. 50 steel for each target temperature, the difference in the stress according to the temperature was not large up to 300 °C, but the stress rapidly decreased above 400 °C. Above 400 °C, the strength of the steel material decreased significantly, while the difference in the room-temperature strength increased with an increase in the strain level. Overall, when the strain level of the maximum stress (tensile strength, $f_u$) is reached, the stress gradually decreases with an increase in the strain. At 800 °C or higher, even if the initial stress was low, the steel did not break, with a continuous increase in the strain of the steel material. The stress–strain curve

for Gr. 50 steel is important for applications of the strength reduction factor according to the strain level in the fireproof design of steel structures.

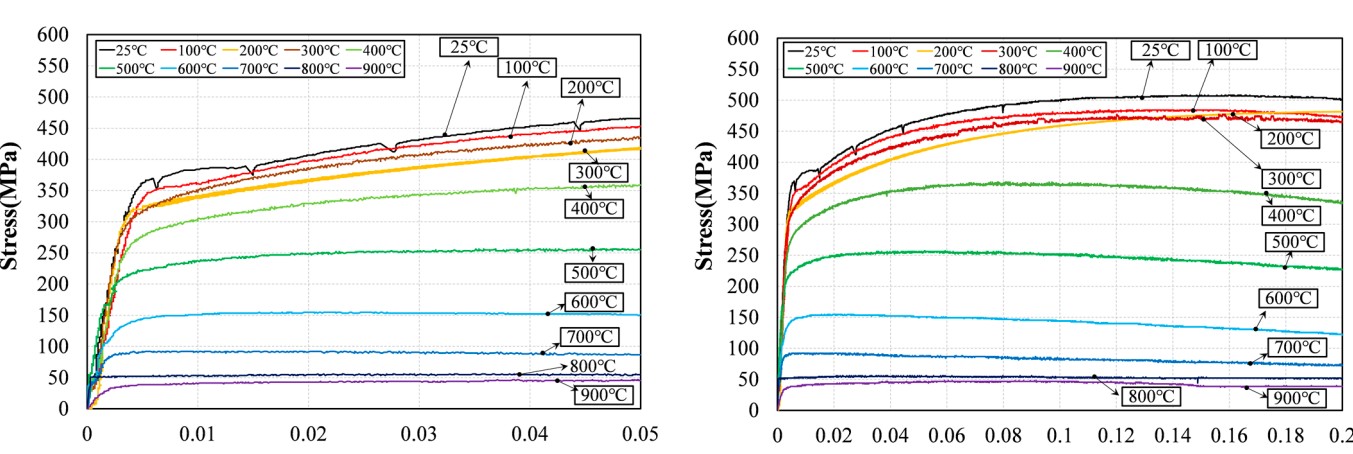

**Figure 6.** Stress–strain curve for Gr. 50 steel: (**a**) 0.05 strain level (initial part); (**b**) 0.2 strain level.

Based on the stress–strain curve shown in Figure 6, the yield strength, tensile strength, and elastic modulus values for the major strain levels are summarized in Table 5. The values in Table 5 were divided by the values at room temperature, which are summarized in the form of a reduction factor of less than 1.0, as listed in Table 6. At 500 °C, the rapid decrease in the reduction factor can be confirmed by the reduction factor, as listed in Table 6. Above 800 °C, the reduced section of the steel rapidly extended and the strain increased, such that the yield strength decreased by 86% and the tensile strength decreased by up to 89%. As the strain level increased from 0.2 to 20%, the strength value for each target temperature gradually increased, but the strength reduction factor decreased at a strain level of 20%. At 300 °C and a strain level of 0.2%, the strength reduction factor decreased by up to 15% compared with room temperature, but at a strain level of 20%, it decreased only by 7%. However, at 400 °C, the strength reduction factor decreased by up to 24% at a strain level of 0.2% compared with room temperature, but decreased by 33% at a strain level of 20%. The reduction factor for the elastic modulus decreased at 100 °C compared with room temperature, increased at 200 °C, and continued to decrease at the target temperature above 300 °C. At 500 °C, the strength reduction factor decreased by up to 41% at a strain level of 0.2% and decreased by 54% at a strain level of 20%, while the elastic modulus reduction factor decreased by 49%. At 600 °C and a strain level of 0.2%, the strength reduction factor decreased by up to 63%, while at a strain level of 20%, it decreased by 76%, where the elastic modulus reduction factor decreased by 60%. Contrary to the results at 300 °C, the strength reduction significantly decreased at the maximum strain rate of 20% for the steel above 400 °C. Other results for the strength reduction factor according to the strain level show that the reduction rate of the strength reduction factor varies with the strain level.

After the test for the ASTM A572 Gr. 60 steel, the stress–strain curves at each target temperature were summarized, as shown in Figure 7. Figure 7a shows the stress–strain curve up to the initial 0.05 (5%) strain level. The stress was measured at temperatures higher than the room-temperature strength due to temperature hardening from 200 to 300 °C. Gr. 60 steels in the room temperature to 300 °C range were more likely to fail at lower strains (6–8%) than steels at other target temperatures due to their brittle behavior, as shown in Figure 7b. However, this type of steel exhibited ductile behavior at a target temperature of 400 °C or higher, with an increase in elongation and the occurrence of ductile fracture. Above 700 °C, the strain increased at low stress, as shown in Figure 7b,

for Gr. 60 steel. At 1000 °C, the deformation of the steel material increased rapidly when the high-temperature extensometer exceeded the limit of the measuring range.

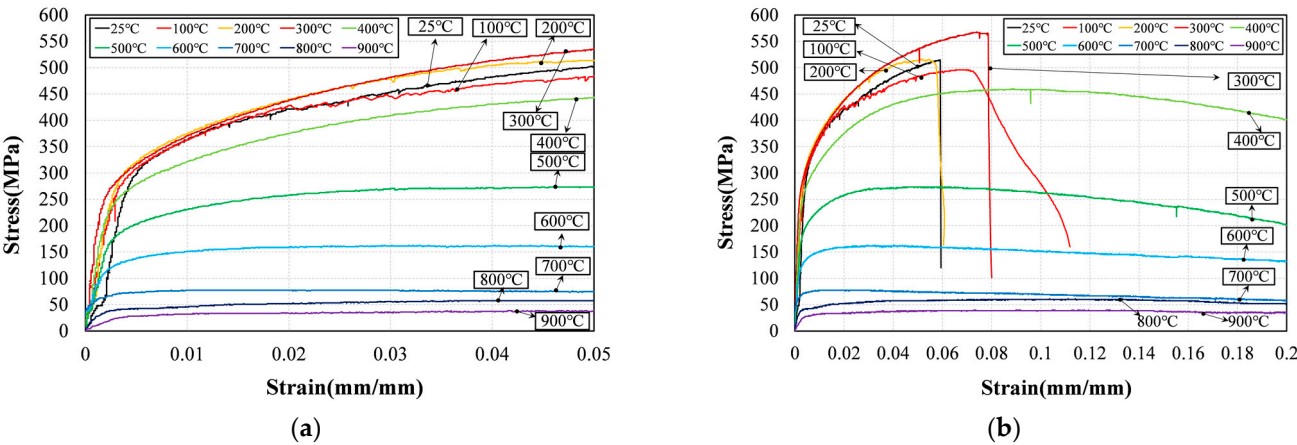

**Figure 7.** Stress–strain curve for Gr. 60 steel: (**a**) 0.05 strain level (initial part); (**b**) 0.2 strain level.

Based on the stress–strain curve in Figure 7, the yield strength, tensile strength, and elastic modulus values for the major strain levels are summarized in Table 7. The values in Table 7 were divided by the values at room temperature, which are summarized in the form of a reduction factor of less than 1.0, as listed in Table 8. At 100 °C, the strength was similar to that at room temperature, but the strength increased in the temperature range from 200 to 300 °C. As the temperature increased above 400 °C, there was a gradual decrease in the strength of the steel. The increase in the strength of the steel in the target temperature range from 200 to 300 °C results in a temperature hardening phenomenon that occurs only in structural steels, which has been reported in several previous studies [31]. As the strain level increased from 0.2% to 20%, there was an increase in strength at each target temperature; however, the strength reduction factor decreased at a strain level of 20%. The strength reduction rate was the smallest at a 0.5% strain level, which was evaluated at a higher strength compared with the other strain levels. The decrease in strength at a 0.2% strain level was up to 10% in the temperature range from 100 to 300 °C, as listed in Table 8; however, at a 0.5% strain level and tensile strength ($f_u / f_{u,25}$), the strength was, at most, 9%, an increase of 10%. This may be due to strain hardening. In addition, in Eurocode 1993-1-2 (Annex A) [9], when the temperature of the steel is less than 400 °C, the tensile strength (ultimate strength) reduction factor is specified to account for strain hardening. At 300 °C, the tensile strength increased by up to 10%; above 400 °C, there was a decrease in the tensile strength. The elastic modulus decreased at a target temperature of 100 to 200 °C, increased again at 300 to 400 °C, and showed a tendency to decrease above 500 °C. At 500 °C, for strain levels of 0.2% and 20%, the strength reduction factor decreased by 41% and 56%, respectively, relative to room temperature, while the elastic modulus reduction factor decreased by 43%. Additionally, at 600 °C, for strain levels of 0.2% and 20%, the strength reduction factor decreased by 61% and 71%, respectively, relative to room temperature, while the elastic modulus reduction factor decreased by 59%. This shows that, similar to the Gr. 50 steel, the strength reduction rate varies with the strain level.

### 3.1.2. Yield Strength

Using the 0.2% offset method, previous studies (Table 2) reporting the strength reduction factor and the test data for ASTM A572 Gr. 50 and 60 were compared, as shown in Figure 8a. The maximum difference in the yield strength reduction between the Gr. 50 (black line) and 60 (red line) steels was 0.1 (10%) at a target temperature of 200 °C. In general, the reduction in the yield strength of the Gr. 50 and 60 steels was similar at the main temperatures of 500 to 600 °C, where there was a rapid decrease in the strength of the steel material. In addition, the values reported in the literature, listed in Table 2, did not meet the

standard (Eurocode, AISC) below 400 °C, as shown in Figure 8a. Therefore, at a 0.2% strain level, for steel materials ranging from room temperature to 400 °C, engineers should consider the value of the yield strength reduction factor. When comparing the NIST regression equation (green dashed line) with the Gr. 50 and 60 test lines, a similar strength reduction factor was obtained up to 400 °C, but differences occurred between 500 and 700 °C. As a result of analyzing the experimental results and the literature (Table 2), the 0.2% strain level yield strength reduction factor should be divided based on the target temperature of 400 °C. At a 0.5% strain level, previous studies (Table 2) on the strength reduction factor and test data for the 0.5% strain level for ASTM A572 Gr. 50 and 60 were compared, as shown in Figure 8b. The graphs of Gr. 50 (black line) and A572 Gr. 50_Kodur (Reg.) below a target temperature of 400 °C show a similar strength reduction even at a strain level of 0.5%; however, a difference occurred between 500 and 700 °C. The strength of Gr. 60 (red line) increased due to temperature strain hardening from 100 to 300 °C. As with the 0.2% strain level, Figure 8b shows that the distribution of data is lower than the reference for a target temperature of 400 °C. The NIST regression equation (green dashed line) and Gr. 50 (black line) showed a similar decrease in the strength below a target temperature of 400 °C, but both were different with respect to Gr. 60 (red line). The Gr. 50 and 60 test lines and EN1993-1-2, AISC, and NIST graphs showed similar strength reductions above a target temperature of 500 °C. Therefore, as the reduction in the yield strength is different based on 400 °C, a separate regression equation should be proposed. In addition, the range of the strain level at 1.0 to 2.0% is the plastic flow region of the steel material, where there is no significant difference because the steel has a constant strength regardless of the strain level (see Figures A2–A4). The regression equations from the literature (Table 2) and the test data for ASTM A572 Gr. 50 and 60 were expanded to a 20% strain level and compared, as shown in Figure 8c. In EN1993-1-2, the stress–strain curve of steel is presented up to 20% (0.2) to obtain a comparison. However, as there are no previous studies analyzing the strength reduction factor at the 20% strain level, the equations were compared using the ASTM A572 Gr. 50 and 60 test data and regression equations from Eurocode, AISC, NIST, and the existing literature (Table 2). As shown in Figure 8c, the strength of the Gr. 60 (red line) steel increased due to temperature strain hardening at a target temperature of 300 °C. Above a target temperature of 400 °C, the strengths of both the ASTM A572 Gr. 50 and 60 steels decreased sharply from 500 °C. In addition, the two steels were significantly lower in strength than the EN1993-1-2, AISC, and NIST. These results show that the strength of steel can be significantly lowered at a high strain level (20%).

As the strain increased, the temperature significantly affected the steel material, resulting in a rapid decrease in the strength. Graphs for Eurocode 1993-1-2 and AISC overestimated the yield strength at all strain levels. Based on the graphs (Figures 8 and A2–A4) of various strain levels, Gr. 60 (red line) had a higher strength than Gr. 50 (black line) below 500 °C, but, above 500 °C, Gr. 60 had a lower strength or similar characteristics compared with Gr. 50. At a target temperature of 400 °C, the HSS of this group, such as Gr. 60, is characterized by increased strength and brittle fracture due to temperature strain hardening. However, above 400 °C, the yield strength of the steel decreased due to the high temperature, which was similar to the graphs for Eurocode 1993-1-2, AISC, NIST, and the experimental data. The results of the previous analysis show that, based on a target temperature of 400 °C, a strength reduction factor regression equation should be separately proposed according to the strength of steel at various strain levels.

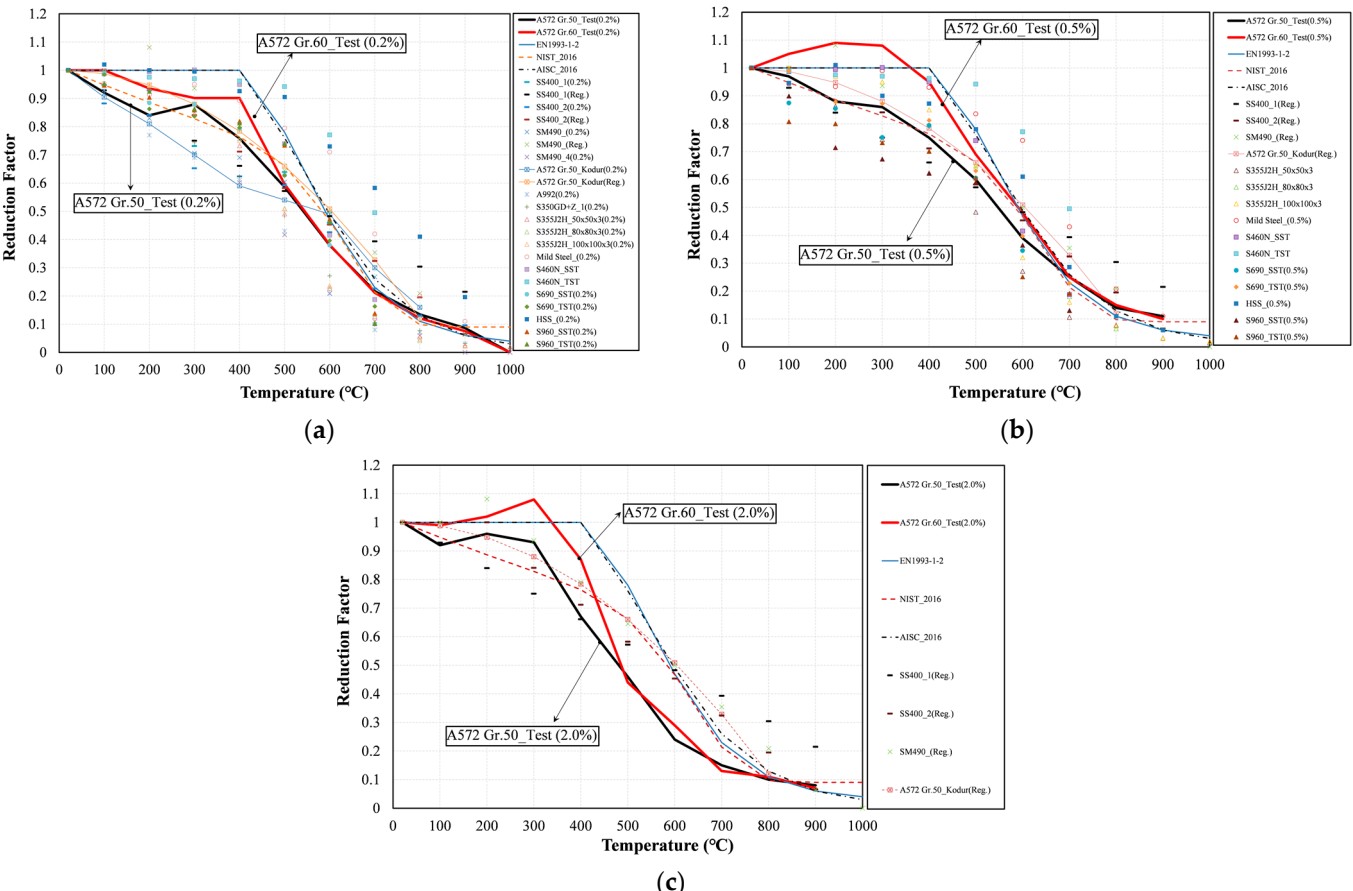

**Figure 8.** Comparison of the reduction factors for the yield strength with the literature values and standards at various strain levels: (**a**) 0.2% (using the 0.2% offset method); (**b**) 0.5%; (**c**) 20%.

### 3.1.3. Elastic Modulus

Using the slope for the elastic section of the steel, Figure 9 compares the reduction factors from the literature and the elastic modulus. The slope of the elastic section refers to the slope of the initial tangent modulus line. The elastic modulus of Gr. 50 (black line) steel increased due to temperature strain hardening at the target temperature of 200 °C. In addition, the elastic modulus of Gr. 60 (red line) steel increased due to temperature strain hardening at a target temperature of 300 °C. The two ASTM A572 Gr. 50 and 60 test lines show significant differences from EN1993-1-2 in the temperature range from room temperature to 400 °C. Above a target temperature of 500 °C, the elastic modulus of both test lines decreased similarly, where the elastic modulus was higher than that of EN1993-1-2 and AISC. At all temperature ranges, there was a significant difference among the NIST graph, EN1993-1-2 graph, and AISC graph. As the distribution of the literature (Table 2) and test data shows that the elastic modulus gradually decreases with an increase in the temperature, a new regression equation for the elastic modulus is proposed in order to reflect these changes.

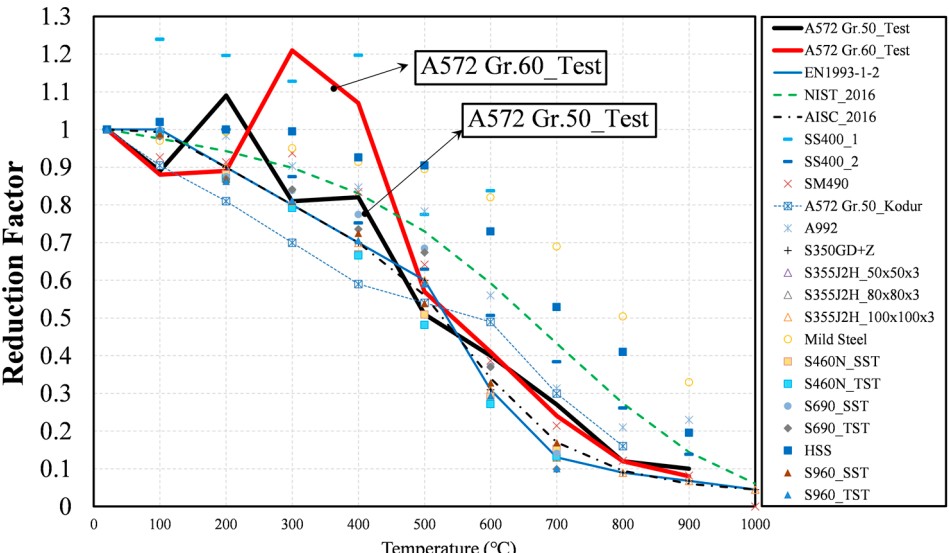

**Figure 9.** A comparison of the reduction factors for the elastic modulus with the literature values and standards.

### 3.1.4. Tensile Strength

The tensile strength reduction factors from the literature (Table 2) and ASTM A572 Gr. 50 and 60 were compared, as shown in Figure 10. In the Eurocode and AISC standards, the tensile strength at the target temperature was divided by the yield strength at room temperature and expressed as a yield ratio. However, in this study, the tensile strength reduction factor was obtained by dividing the tensile strength at the target temperature by the tensile strength at room temperature ($f_u / f_{u,25}$). To compare the tensile strength reduction factor based on an equal normalization, the literature values (Table 2) or codes presenting yield ratios ($f_u / f_y$) were changed to the tensile strength ratio ($f_u / f_{u,25}$).

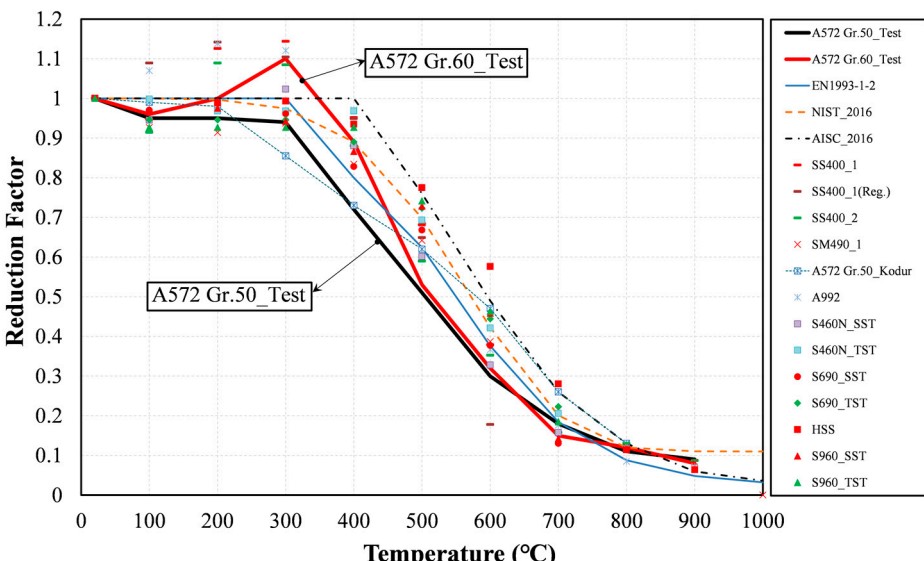

**Figure 10.** A comparison of the reduction factors for the tensile strength with the literature values and standards.

Tables 6 and 8 list the tensile strength reduction factors ($f_u / f_{u,25}$) of ASTM A572 Gr. 50 and 60 steels. The tensile strength of the steel was the highest value at all strain levels. The tensile strength of Gr. 60 steel (red line) increased by ~10% at 300 °C due to temperature strain hardening. However, the tensile strength reduction factor of Gr. 50

(black line) was observed to be 0.28 (28%), which is different from the AISC graph. The two ASTM A572 Gr. 50 and 60 test lines showed significant differences in the target temperature range from 300 to 400 °C, as well as a similar strength reduction above 500 °C. Above a target temperature of 400 °C, the Gr. 50 line was located below the distribution of the other data. In addition, the distribution of the data was below the intensity reduction graph for the AISC at all target temperatures. There was no significant decrease in the tensile strength from room temperature to 300 °C, but the decrease in the tensile strength increased significantly above 400 °C. Therefore, a new tensile strength reduction factor regression equation was proposed based on the target temperature of 300 °C.

### 3.2. Proposed Reduction Factors for the Strength and Elastic Modulus

The proposed equations for the yield strength, tensile strength, and elastic modulus reduction factors were derived from data reported in previous studies [9,13,16–19,21–27,31–37] and the ASTM A572 test results. In the following equations, $T_a$ is the temperature of steel (°C), $R_{f,Y}$ is the yield strength reduction factor of the steel in the regression equation, $R_{f,E}$ is the elastic modulus reduction factor of the steel in the regression equation, and $R_{f,T}$ is the tensile strength reduction factor of the steel in the regression equation. MS, including Gr. 50, is represented by the black line, while HSS, including Gr. 60, is shown by the red line. The non-linear least squares regression was used. To include both the experimental and literature data, various factors were considered. The average values of all of the test data were used to derive the proposed equation, where the numerical value was adjusted to simplify the equation.

### 3.2.1. Yield Strength (0.2% Strain Level)

As shown in Figure 11, the MS strength reduction factors are always lower than those of HSS with an increase in the temperature. In previous studies, HSS was shown to display a characteristic vulnerability to temperature in yield strength compared to MS [13,31]. However, using the average value of the data, the regression equation showed that MS was more vulnerable to temperature than HSS, in contrast to Figure 1. Compared to the HSS, the yield strength of the MS decreased significantly with the increasing temperature. Thus, the maximum difference in the yield strength resulted by 17% at 400 °C particularly. Therefore, we added the temperature range at 400 °C in the regression equation. To reflect these observations, the yield strength reduction factor for MS is summarized in Equations (1)–(3), whereas the yield strength reduction factor for HSS is summarized in Equations (4)–(6). The yield strengths obtained from the experiments at various strains are insufficient to present the yield strength reduction factor for each steel grade according to the strain level. To elucidate the equations for various strain levels, future studies must analyze the yield strength at additional strain levels. Therefore, in this study, the following yield strength reduction factors are proposed by considering the variables at a 0.2% strain level.

- MS (Mild Steel): Yield Strength Reduction Factor Equation

$$\text{For } 25\,°\text{C} \leq T_a \leq 400\,°\text{C}:$$
$$R_{f,Y} = -2 \times 10^{-7}(T_a)^2 - 1 \times 10^{-4}(T_a) + 1. \tag{1}$$

$$\text{For } 400\,°\text{C} < T_a \leq 800\,°\text{C}:$$
$$R_{f,Y} = 2 \times 10^{-6}(T_a)^2 - 4.1 \times 10^{-3}(T_a) + 2.267. \tag{2}$$

$$\text{For } 800\,°\text{C} < T_a \leq 1000\,°\text{C}:$$
$$R_{f,Y} = 3 \times 10^{-8}(T_a)^2 - 1 \times 10^{-3}(T_a) + 1. \tag{3}$$

- HSS (High-Strength Steel): Yield Strength Reduction Factor Equation

$$\text{For } 25\,°\text{C} \leq T_a \leq 400\,°\text{C}:$$
$$R_{f,Y} = -1 \times 10^{-6}(T_a)^2 - 2 \times 10^{-4}(T_a) + 1. \tag{4}$$

For $400\ °C < T_a \le 800\ °C$ :
$$R_{f,Y} = 7 \times 10^{-7}(T_a)^2 - 2.4 \times 10^{-3}(T_a) + 1.6008. \tag{5}$$

For $800\ °C < T_a \le 1000\ °C$ :
$$R_{f,Y} = 5 \times 10^{-7}(T_a)^2 - 1.5 \times 10^{-3}(T_a) + 1. \tag{6}$$

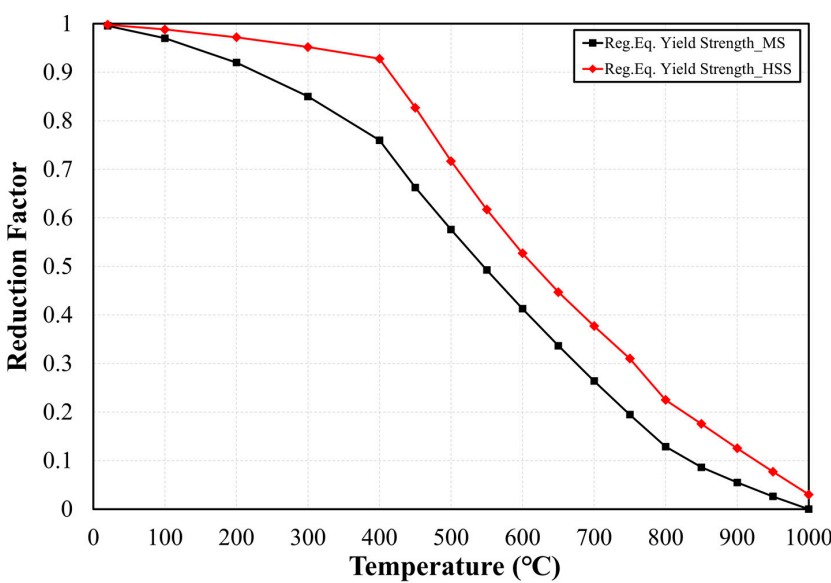

**Figure 11.** Proposed reduction factor for the yield strength.

### 3.2.2. Elastic Modulus

The elastic modulus reduction factor showed differences according to the MS and HSS zones. The elastic modulus of HSS was low as the temperature increased compared with MS. Figure 12 shows the proposed equation for the elastic modulus reduction factor for MS and HSS. The elastic modulus is an important factor that determines the initial strength, expressed as the tangent modulus in the elastic section of the stress–strain diagram. Compared with MS, the reduction factor of the elastic modulus for HSS is lower with an increase in the temperature. The elastic modulus reduction factor for MS is expressed in Equation (7), while that for HSS is expressed in Equation (8).

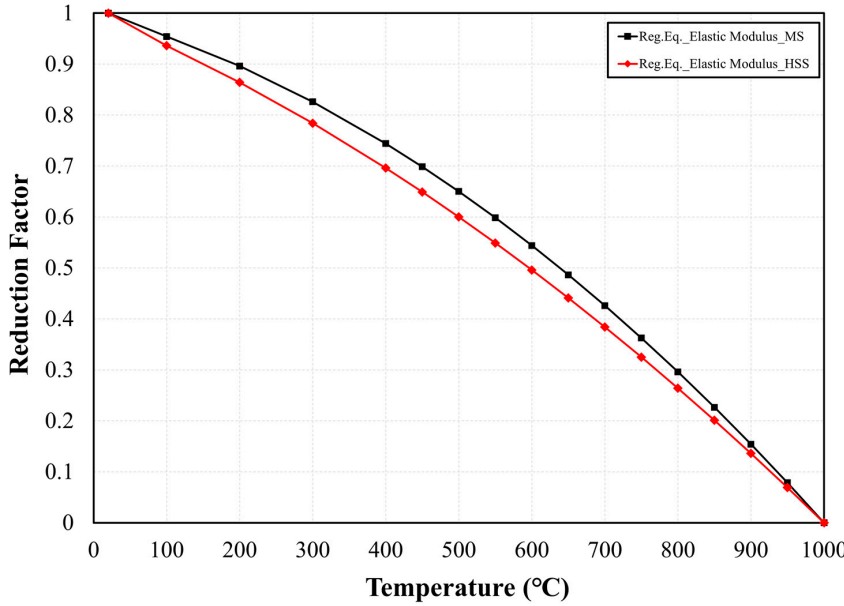

**Figure 12.** Proposed reduction factor for the elastic modulus.

- MS (Mild Steel): Elastic Modulus Reduction Factor Equation

$$\text{For } 25\,^{\circ}\text{C} \leq T_a \leq 1000\,^{\circ}\text{C}:$$
$$R_{f,E} = -6 \times 10^{-7}(T_a)^2 - 4 \times 10^{-4}(T_a) + 1. \tag{7}$$

- HSS (High-Strength Steel): Elastic Modulus Reduction Factor Equation

$$\text{For } 25\,^{\circ}\text{C} \leq T_a \leq 1000\,^{\circ}\text{C}:$$
$$R_{f,E} = -4 \times 10^{-7}(T_a)^2 - 6 \times 10^{-4}(T_a) + 1. \tag{8}$$

### 3.2.3. Tensile Strength

Figure 13 shows the proposed equation for the tensile strength reduction of MS and HSS. The difference in the tensile strength reduction factor was not significant, regardless of the steel grade. As the tensile strength shows the greatest strength in the stress–strain curve, there is no difference depending on the strain level. The tensile strength reduction factor for MS is summarized by Equations (9)–(11), while that for HSS is expressed in Equations (12)–(14). The difference between the reduction factors of MS and HSS is not large, such that it is not necessary to divide them into regression equations. However, if a specific fire design is required above 500 °C, the equations can be applied separately.

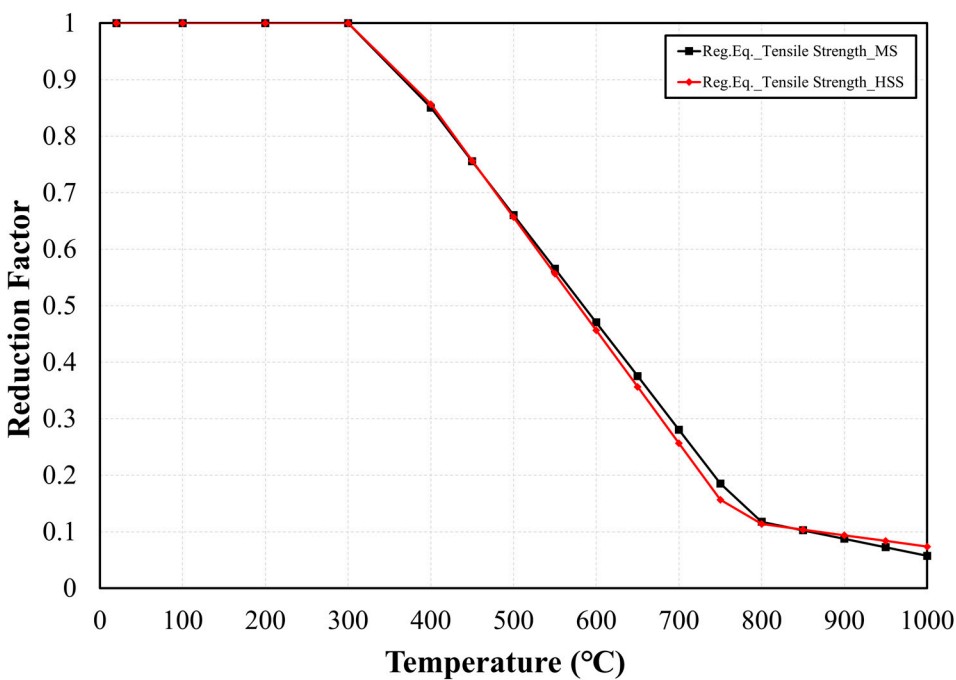

**Figure 13.** Proposed reduction factor for the tensile strength.

- MS (Mild Steel): Tensile Strength Reduction Factor Equation

$$\text{For } 25\,^{\circ}\text{C} \leq T_a \leq 300\,^{\circ}\text{C}:$$
$$R_{f,T} = 1. \tag{9}$$

$$\text{For } 300\,^{\circ}\text{C} < T_a \leq 800\,^{\circ}\text{C}:$$
$$R_{f,T} = -1.9 \times 10^{-3}(T_a) + 1.6104. \tag{10}$$

$$\text{For } 800\,^{\circ}\text{C} < T_a \leq 1000\,^{\circ}\text{C}:$$
$$R_{f,T} = -3 \times 10^{-4}(T_a) + 0.3575. \tag{11}$$

- HSS (High-Strength Steel): Tensile Strength Reduction Factor Equation

$$\text{For } 25\,°C \le T_a \le 300\,°C:$$
$$R_{f,T} = 1.$$

(12)

$$\text{For } 300\,°C < T_a \le 800\,°C:$$
$$R_{f,T} = -2 \times 10^{-3}(T_a) + 1.6565.$$
$$\left( \text{and } R_{f,T} = 0.86 \text{ at } 400\,°C, \ R_{f,T} = 0.26 \text{ at } 700\,°C \right)$$

(13)

$$\text{For } 800\,°C < T_a \le 1000\,°C:$$
$$R_{f,T} = -2 \times 10^{-4}(T_a) + 0.2738.$$

(14)

### 3.3. Comparisons of the Proposed Equations with the Literature

In this section, the proposed equations for the yield strength, tensile strength, and elastic modulus are compared to the literature (Table 2) values and reference data. In Figures 14–16, the black bold solid line is the proposed equation for MS (Equation), whereas the red bold solid line is the proposed equation for HSS (Equation). The black dashed line shows the test results for Gr. 50, whereas the red dashed line shows the test results for Gr. 60.

The values in Tables 9–11 were converted by the reduction values from Figures 14–16, respectively. Tables 9–11 are divided into Column (a)–(f) and Row (1), (2) for convenience. For example, when comparing the yield strength of the Eurocode (E) and MS equation (M) in Table 9, the difference is 24%, which can be found in Column (a) and Row (1) (Table 9, (a)(1)).

**Table 9.** Comparison yield strength value (%) of standard and equation at main temperature.

| Equation | Temp. | 400 °C | | | 700 °C | | | Row |
|---|---|---|---|---|---|---|---|---|
| | Standard | EC *, (E) | AISC, (A) | NIST, (N) | EC *, (E) | AISC, (A) | NIST, (N) | |
| MS Equation (Equations (1)–(3), M) | Difference (%) | $|M-E|$ | $|M-A|$ | $|M-N|$ | $|M-E|$ | (M)–(A) | (M)–(N) | |
| | | 24 | 24 | 0 | 3 | 0 | 5 | (1) |
| HSS Equation (Equations (4)–(6), H) | Difference (%) | $|H-E|$ | $|H-A|$ | $|H-N|$ | $|H-E|$ | (H)–(A) | (H)–(N) | |
| | | 7 | 7 | 16 | 15 | 12 | 16 | (2) |
| | Column | (a) | (b) | (c) | (d) | (e) | (f) | |

\* EC: Eurocode.

**Table 10.** Comparison elastic modulus value (%) of standard and equation at 700 °C.

| Equation | Temp. | 700 °C | | | Row |
|---|---|---|---|---|---|
| | Standard | EC * (E) | AISC (A) | NIST (N) | |
| MS Equation (Equation (7), M) | Difference (%) | $|M-E|$ | $|M-A|$ | $|M-N|$ | |
| | | 30 | 26 | 0 | (1) |
| HSS Equation (Equation (8), H) | Difference (%) | $|H-E|$ | $|H-A|$ | $|H-N|$ | |
| | | 25 | 21 | 5 | (2) |
| | Column | (a) | (b) | (c) | |

\* EC: Eurocode.

**Table 11.** Comparison tensile strength value (%) of standard and equation at main temperature.

| Equation | Temp. | 400 °C | | | 700 °C | | | Row |
|---|---|---|---|---|---|---|---|---|
| | Standard | EC * (E) | AISC (A) | NIST (N) | EC * (E) | AISC (A) | NIST (N) | |
| MS Equation (Equations (9)–(11), M) | Difference (%) | $|M-E|$ | $|M-A|$ | $|M-N|$ | $|M-E|$ | $|M-A|$ | $|M-N|$ | |
| | | 5 | 15 | 4 | 10 | 2 | 8 | (1) |
| HSS Equation (Equations (12)–(14), H) | Difference (%) | $|H-E|$ | $|H-A|$ | $|H-N|$ | $|H-E|$ | $|H-A|$ | $|H-N|$ | |
| | | 6 | 14 | 3 | 8 | 0 | 6 | (2) |
| | Column | (a) | (b) | (c) | (d) | (e) | (f) | |

\* EC: Eurocode.

Figure 14 shows the differences in the yield strength between the tests performed here and other tests from literature surveys (Table 2) at the 0.2% offset strain level. At the 0.2% strain level, the normalized strength factor of MS is lower than that of HSS, which is different from the results of Chen et al. [13,31]. The yield strength was compared with the proposed equation, Equations (1)–(6), and standards (EN 1993-1-2 and AISC 2016). The yield strength results of the compared standards and equation at the main temperatures (400, 700 °C) are summarized in Table 9 from Figure 14. The differences in the reduction factor between MS (Equation (1)) the and standards (EN1993-1-2 and AISC 2016) are all 24% at 400 °C (Table 9, (a)(1), (b)(1)). Moreover, the HSS ((Equation (4)) and standards (EN1993-1-2 and AISC 2016) are all 7% at 400 °C (Table 9, (a)(2), (b)(2)). At 700 °C, the difference in yield strength reduction factor between MS (Equation (2)) and EN1993-1-2 is 3% (Table 9, (d)(1)), and the difference between MS (Equation (2)) and AISC 2016 is 0 (Table 9, (e)(1)). In addition, the difference between HSS (Equation (5)) and EN1993-1-2 is 15% (Table 9, (d)(2)), and the difference between HSS (Equation (5)) and AISC 2016 is 12% (Table 9, (e)(2)). NIST 1907 (2016) was compared with the proposed yield strength equation (Equations (1)–(6)). At 400 °C, the difference between the yield strength reduction factor of MS (Equation (1)) and NIST was 0% (Table 9, (c)(1)), and the difference between HSS (Equation (4)) and NIST was 16% (Table 9, (c)(2)). At 700 °C, the difference in the yield strength reduction factor between MS (Equation (1)) and NIST was 5% (Table 9, (f)(1)), and the difference between HSS (Equation (4)) and NIST was 16% (Table 9, (f)(2)). These results mean that HSS (Equations (4)–(6)) is closer to the yield strength reduction factor of Eurocode and AISC than MS (Equations (1)–(3)) at 400 °C or below. Previously, Eurocode and AISC were known to reflect the yield strength reduction factor of the MS group, but the results were different from the actual data. The regression equation of NIST 1907 (2016) is closer to the yield strength reduction factor of MS (Equations (1)–(3)) than HSS (Equations (4)–(6)) at 400 °C or below. Because the NIST's regression equation was created based on experimental data from various mild steels, the results were relatively similar to those of the MS (Equations (1)–(3)) equation.

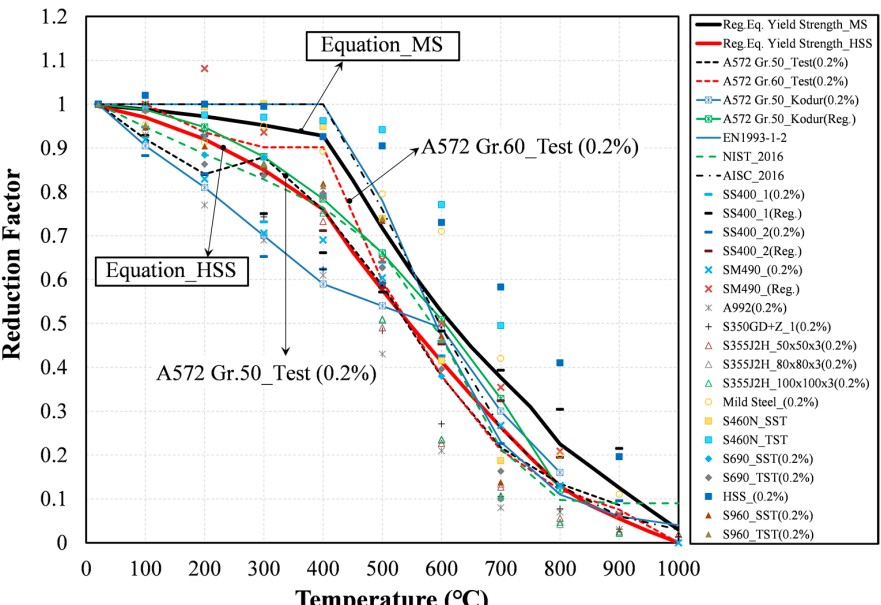

**Figure 14.** A comparison of the reduction factors for the yield strength with the literature values and standards at a 0.2% strain level.

Figure 15 shows the regression equation for the proposed elastic modulus reduction factor with a comparison to the literature (Table 2) and code values. The proposed equation is higher than the reduction factor for the elastic modulus for EN1993-1-2 and AISC because the distribution of the literature data is generally higher than the code values

(i.e., EN1993-1-2 and AISC). The proposed equation for the elastic modulus (Equations (7) and (8)) and the standards (Eurocode and AISC) were compared. The elastic modulus results of the compared standards and equation at 700 °C are summarized in Table 10 from Figure 15. At 700 °C, the difference in the elastic modulus reduction between MS (Equation (7)) and EN 1993-1-2 is 30% (Table 10, (a)(1)), and the difference between HSS (Equation (8)) and EN 1993-1-2 is 25% (Table 10, (a)(2)). At 700 °C, the difference in the elastic modulus reduction between MS (Equation (7)) and AISC 2016 is 26% (Table 10, (b)(1)), and the difference between HSS (Equation (8)) and AISC 2016 is 21% (Table 10, (b)(2)). On the other hand, at 700 °C, there was no difference in the elastic modulus reduction between MS (Equation (7)) and NIST (0%), and the difference between HSS (Equation (8)) and NIST is 5% (Table 10, (c)(1), (c)(2)).

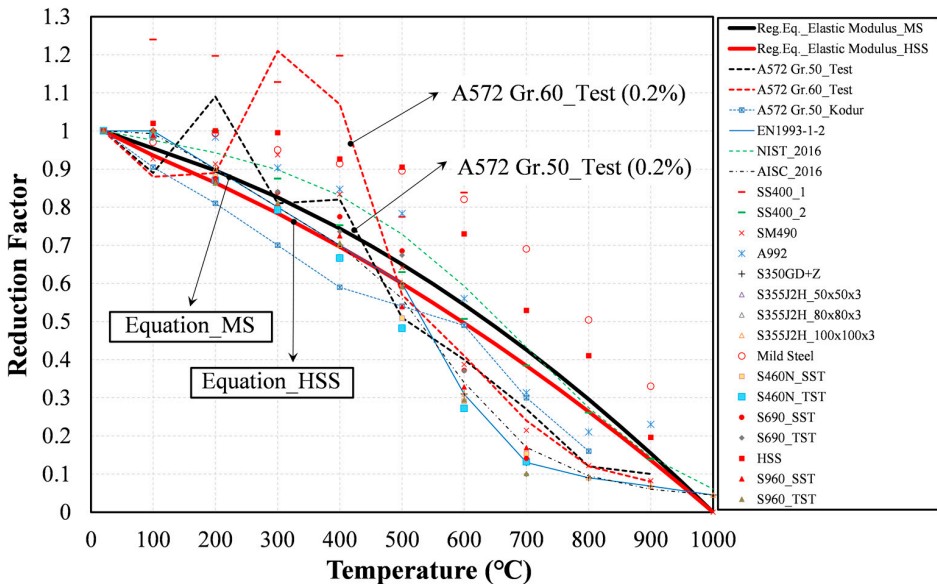

**Figure 15.** A comparison of the reduction factors for the elastic modulus with the literature values and standards.

As with the yield strength results, the proposed elastic modulus equation (Equations (7) and (8)) was found to be closer to NIST compared to Eurocode or AISC. The difference between the proposed equation and Eurocode and AISC is noteworthy because the proposed equation of the elastic modulus reduction factor in this study includes the extensive experimental data of the previous study. Therefore, more research is required to derive the elastic modulus equation according to the steel grade.

Figure 16 shows the regression equation for the proposed tensile strength reduction factor with a comparison to the literature (Table 2) and code values. The proposed equation graph is higher than EN1993-1-2 and tends to be similar to NIST. Above 300 °C, the proposed graph underestimates the tensile strength factor, as compared with AISC. The proposed equation for tensile strength (Equations (9)–(14)) and standards (Eurocode, AISC) were compared. The tensile strength result of the compared standards and equation at the main temperatures (400 °C, 700 °C) are summarized in Table 11, in which the values were converted by the reduction values from Figure 16. At 300 °C, there was no difference (0%) between the standards (Eurocode, AISC) and the proposed equation (Equations (9)–(14)) in the tensile strength reduction factor. However, at 400 °C, the difference in the tensile strength reduction factor between MS (Equations (9)–(11)) and EN1993-1-2 is 5% (Table 11, (a)(1)), and the difference between HSS (Equations (12)–(14)) and EN1993-1-2 is 6% (Table 11, (a)(2)). At 400 °C, the difference in the tensile strength reduction factor between MS (Equations (9)–(11)) and AISC 2016 is 15% (Table 11, (b)(1)), and the difference between HSS (Equations (12)–(14)) and AISC 2016 is 14% (Table 11, (b)(2)). At 700 °C, the difference in the tensile strength reduction factor between MS (Equations (9)–(11)) and

EN1993-1-2 is 10% (Table 11, (d)(1)), and the difference between HSS (Equations (12)–(14)) and EN1993-1-2 is 8% (Table 11, (d)(2)). At 700 °C, the difference in the tensile strength reduction factor between MS (Equations (9)–(11)) and AISC (2016) is 2% (Table 11, (e)(1)), and the difference between HSS (Equations (12)–(14)) and AISC (2016) is 0% (Table 11, (a)(2)). The difference between the proposed MS and HSS equations (Equations (9)–(14)) and the tensile strength reduction factor of NIST (2016) is 3% at 300 °C. At 400 °C, the difference in the tensile strength reduction factor between MS (Equations (9)–(11)) and NIST is 4% (Table 11, (c)(1)), and the difference between HSS (Equations (12)–(14)) and NIST is 3% (Table 11, (c)(2)). At 700 °C, the difference in the tensile strength reduction factor between MS (Equations (9)–(11)) and NIST is 8% (Table 11, (f)(1)), and the difference between HSS (Equations (12)–(14)) and NIST (2016) is 6% (Table 11, (f)(2)). Tensile strength showed no significant difference depending on the steel type compared to the yield strength and elastic modulus. However, since the largest difference occurred between the standards (EN 1993-1-2, AISC) and the data in the 300 ~ 700 °C range, a new equation was proposed at 300 °C and 700 °C.

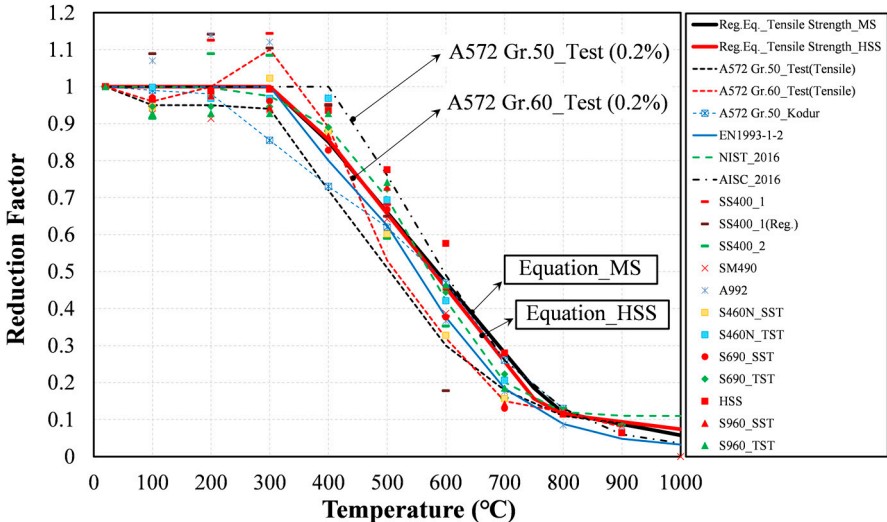

**Figure 16.** A comparison of the reduction factors for the tensile strength with the literature values and standards.

Unlike previous studies, this study proposes an equation divided into mild steel (MS) and high-strength steel (HSS). It is necessary to distinguish the yield strength according to the steel type. Through further tests and studies, it is necessary to derive the yield strength reduction factor equation at different strain rates. An approach similar to this study can help to establish a fire-resistant design standard for each steel type in the future.

## 4. Conclusions

This study presents a detailed experimental study on the mechanical properties of ASTM A572 Grade 50 (Gr. 50) and Grade 60 (Gr. 60) steels at high temperatures. In addition, extensive literature data on the yield strength, tensile strength, and elastic modulus of mild steel (MS), high-strength steel (HSS), and very-high-strength steel (VHSS) are provided. Both ASTM A572 Gr. 50 and 60 steels were used in the experimental tests under steady-state fire conditions. The yield strength reduction factor was analyzed by expanding the strain up to 20%. Based on the test and existing literature data, regression equations for yield strength, tensile strength, and elastic modulus were proposed. The test data and the regression equation were compared with the reduction factors suggested by the standards, including Eurocode, AISC, and NIST. Based on the results of this study, the following conclusions can be drawn:

- When comparing the proposed MS Equations (1)–(3) of the yield strength reduction with the standard (EN 1993-1-2 and AISC 2016), the maximum difference of 24%

occurred at 400 °C. On the other hand, the proposed MS Equations (1)–(3) did not differ from the ones of the NIST 1907 (0%). This indicates that the proposed MS Equations (1)–(3) and NIST equations differ from the ones of the standards (EN 1993-1-2 and AISC 2016) based on experimental data of various mild steels.

- Eurocode and AISC propose a yield strength reduction factor of 1.0 for mild steel at 400 °C, but the proposed MS Equations (1)–(3) and the NIST equations indicate a value of 0.76. Although the point at which the strength of steel material rapidly decreases is known to be 538 °C, this study suggests that the yield strength reduction starts from 400 °C for MS.
- The proposed Equations (4)–(6) of the yield strength reduction for the HSS at 400 °C are different from the ones of the standard (EN 1993-1-2, AISC 2016) by 7%, and differ by 16% from the NIST. The MS Equations (1)–(3) showed a difference of 5% from the ones of the NIST equation at 700 °C. The HSS Equations (4)–(6) were 16% different from the ones of the NIST at 700 °C.
- These results indicate that the HSS Equations (4)–(6) are closer to the yield strength reduction factor suggested by the standards (Eurocode and AISC) than the MS Equations (1)–(3) at 400 °C. However, it does not work at 700 °C such that the yield strength reduction factor of the HSS increases greatly compared with the standards (Eurocode and AISC).
- In the case of HSS, it is better to make the yield strength reduction factors lower than the ones of current design standards to ensure a safe design. Moreover, the strength reduction factors of the MS at both 400 °C and 700 °C agreed well with the NIST code rather than the standards, including the Eurocode and AISC.
- The differences in the elastic modulus reduction factor of the Eurocode and AISC to MS are 30% and 26%, respectively. However, the elastic modulus factors of the NIST are in good agreement under 5% based on the experimental works for both MS and HS. In the case of the elastic modulus reduction factor for the HSS, the decrement difference was approximately 25% from the one of Eurocode. As with the yield strength results, the proposed elastic modulus Equations (7) and (8) were close to the ones of the NIST compared to Eurocode or AISC.
- As a result of analyzing the tensile strength, there was a negligible difference in the reduction factor depending on the steel type compared to the yield strength and the elastic modulus. Because the minor difference occurred in the 300~700 °C range, it is necessary to perform careful investigation for much higher strength levels of steel.
- The reduction factor of the yield strength between MS and HSS showed consistently decaying patterns throughout all temperature ranges. The yield strength reduction of the HSS was smaller by 17% than the one of MS at 400 °C. Generally speaking, the yield strength, elastic modulus, and tensile strength equations of NIST 1907 (2016) are prepared based on experimental data for various mild steels, and the result is relatively similar to the equation proposed in this study.
- This paper covered various strength reduction factors, including the yield strength, tensile strength, and elastic modulus factors at elevated temperatures of up to 1000 °C. This paper only covers the comparisons of the strength reduction effects of the Grade 50 and 60 structural steels to some design codes, including EN, AISC and NIST. Further developments and comparisons on the strength reduction factors using various levels of high-strength structural steels are needed in the future. Moreover, the creep effects at elevated temperatures were not covered in this paper due to limitations. Future work needs to cover the creep effect at elevated temperatures with various stress levels in the fire-resistant design.

**Author Contributions:** Conceptualization, S.-H.L. and B.-J.C.; methodology, S.-H.L. and B.-J.C.; formal analysis, S.-H.L. and B.-J.C.; investigation, S.-H.L. and B.-J.C.; resources, S.-H.L. and B.-J.C.; data curation, S.-H.L. and B.-J.C.; writing—original draft preparation, S.-H.L. and B.-J.C.; writing—review and editing, S.-H.L. and B.-J.C.; visualization, S.-H.L.; supervision, B.-J.C. All authors have read and agreed to the published version of the manuscript.

**Funding:** This research was funded by Kyonggi University's Graduate Research Assistantship 2018, grant number 20181102071, and the Korea Institute of Energy Technology Evaluation and Planning (KETEP) and the Ministry of Trade, Industry & Energy (MOTIE) of the Republic of Korea, grant number 20161510400110.

**Institutional Review Board Statement:** Not applicable.

**Informed Consent Statement:** Not applicable.

**Data Availability Statement:** Data are contained within the article.

**Acknowledgments:** This work was supported by Kyonggi University's Graduate Research Assistantship 2018 and the Korea Institute of Energy Technology Evaluation and Planning (KETEP) and the Ministry of Trade, Industry & Energy (MOTIE) of the Republic of Korea (No. 20161510400110).

**Conflicts of Interest:** The authors declare no conflict of interest.

## Appendix A

**Table A1.** Chemical composition of ASTM A572 Grade 50 and 60 steels.

| Chemical Composition | Steel Type | |
|---|---|---|
| | **ASTM A572 Grade 50 (Gr. 50)** | **ASTM A572 Grade 60 (Gr. 60)** |
| Carbon, C (%) | 0.23 | 0.26 |
| Iron, Fe (%) | 98 | 98 |
| Manganese, Mn (%) | 1.35 | |
| Phosphorus, P (%) | 0.04 | |
| Silicon, Si (%) | 0.40 | |
| Sulfur, S (%) | 0.05 | |

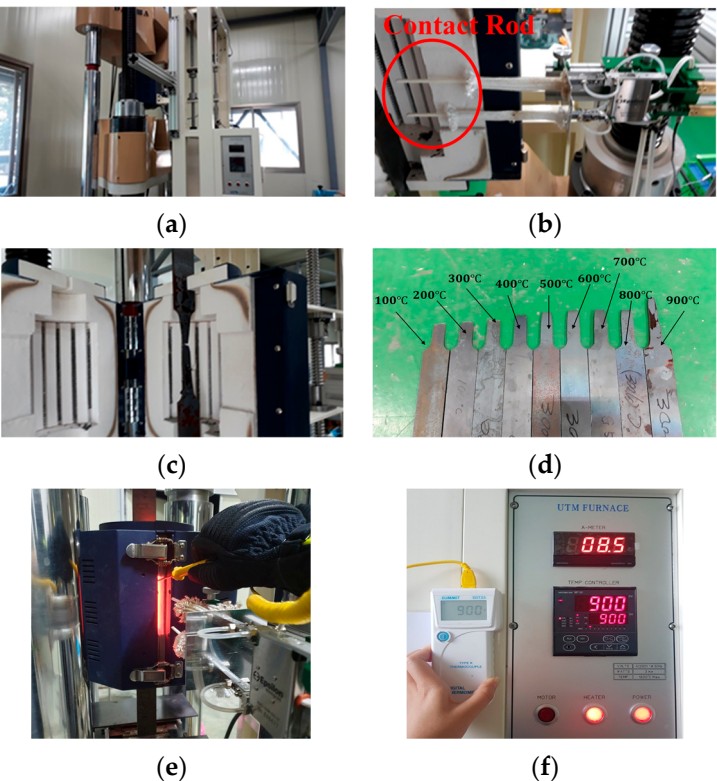

(a)

(b)

(c)

(d)

(e)

(f)

**Figure A1.** Equipment for the steady-state test (SST): (**a**) heating furnace; (**b**) extensometer; (**c**) steel after the high-temperature tensile test; (**d**) Gr. 50 specimens after the high-temperature tensile test; (**e**) high-temperature tensile test using the extensometer and thermocouple; (**f**) steel temperature (surface temperature and furnace temperature).

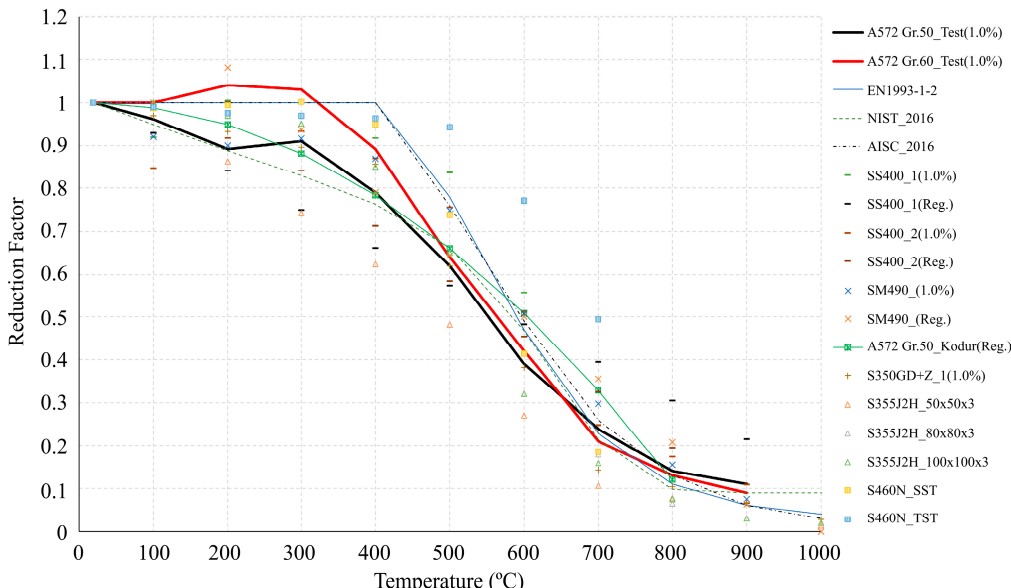

**Figure A2.** Comparison of reduction factors for yield strength with previously reported values and standards at 1.0% strain level.

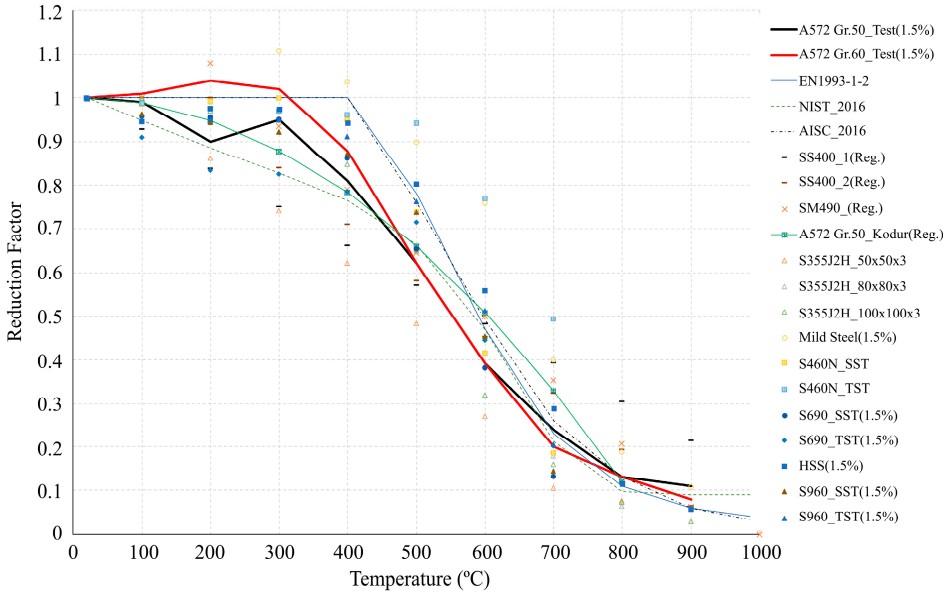

**Figure A3.** Comparison of reduction factors for yield strength with previously reported values and standards at 1.5% strain level.

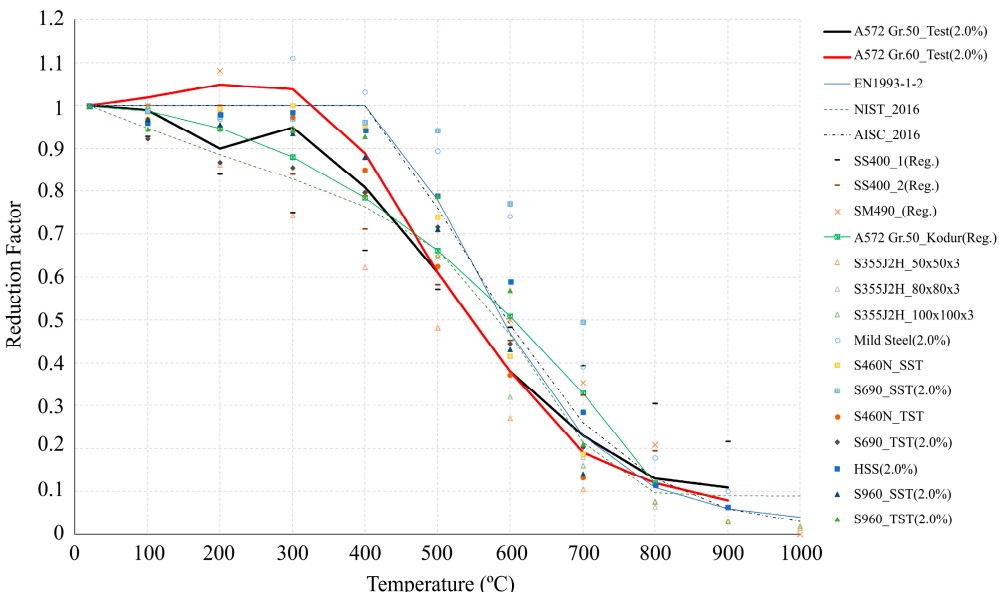

**Figure A4.** Comparison of reduction factors for yield strength with previously reported values and standards at 2.0% strain level.

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
