# Peer review of "Mechanical Properties of ASTM A572 Grades 50 and 60 Steels at High Temperatures"

_applsci, doi:10.3390/app112411833_

Round 1

Reviewer 1 Report

Dear Authors, 

The work presents tests of the mechanical properties of ASTM A572 Grades 50 and 60 Steels 2 at high temperatures. 
An extensive literature review has been carried out, which adequately introduces the issue. The article has been divided into appropriate sections.
Select the appropriate and current literature.
The presented research results are consistent.
Some figures need to be corrected.
No markings are visible in Figures 6, 7, 8, 14, 15, 16. It is worth correcting this to make the charts more readable.

Author Response

Thank you for taking the time to review the manuscript.

I checked the comments given to the author and amended as attachment.

Content revised in the manuscript due to the comments of reviewers is indicated in red letters.

Reviewer 2 Report

The paper analyzed the effect that elevated temperatures have on the mechanical properties of ASTM A572 Gr. 50 and 60 steels. To this end, extensive literature data on yield strength, tensile strength, and elastic modulus of mild steel(MS), high strength steel(HSS) and very high strength steel (VHSS) were described. The authors present a detailed experimental study on the mechanical properties of ASTM A572 Grade 50 (Gr. 50) and Grade 60 (Gr. 60) steels at high temperatures. Furthermore, equations for the yield strength, tensile strength, and elastic modulus were proposed. The title and abstract are appropriate for the content of the text. Furthermore, the article is well constructed, the experiments were well conducted, and analysis was well performed.

Only 2 areas need revision:

  1. Page 12, para 345, para347: There are 2 however, it is recommended to delete or use other words.
  2. 1-3, Fig.6-7: The quality of the picture is too low, the temperature figure in the picture is not clear.

Author Response

(The authors gave the same response as above.)

Reviewer 3 Report

The authors presented research on the mechanical properties of HSS at elevated temperatures. The presented results may be interesting for potential readers due to the possibility of implementing the obtained results. However, the structure of the manuscript looks more like an industrial report than a research article. The tests were performed without deeper interpretation (considering microstructure) of the results and comparison with the literature. The thermal stability of HHS and short term soaking at elevated temperatures can be compared to short tempering. The subject of the tempering of steels and the reduction of certain mechanical properties is well known. Mainly, it depends on material structure and heat treatment methods, which was not presented. 

Other remarks:

  1. The chemical composition and hardness should be provided.
  2. The microstruture and heat treatment description should be provided.
  3. Figures 1, 2, 3, 6 and 7 are difficult to read.
  4. The discussion should be extended to include the influence of microstructure, which is described in the literature.

Author Response

(The authors gave the same response as above.)

Round 2

Reviewer 3 Report

The authors' responses are exhaustive. Nevertheless, I believe that the microstructure is important as it has a significant influence on the mechanical properties and thermal stability. Mechanical testing is important, but extending them to include material issues could explain the changes in their thermal stability. Such investigations would increase the value of this manuscript.The authors explained to conduct such studies in the future. After the author's explanations, I believe that the article is also interesting for potential readers in the current scope of research, and its results can be used as material data.